# From Reasoning Traces to Reusable Modules: Understanding Compositional Generalization in Language Model Reasoning

Lingjing Kong [1] [*]   Xin Liu [2] [*]   Guangyi Chen [2] [1]   Martin Q. Ma [1]   Xiangchen Song [1]   Yuekai Sun [3] [4]
Mikhail Yurochkin [3]   Taylor W. Killian [3]   Ruslan Salakhutdinov [1]   Kun Zhang [1] [2]   Eric P. Xing [1] [2] [3]
Zhengzhong Liu [3]

## Abstract

Post-training pipelines that combine supervised fine-tuning (SFT) with reinforcement learning (RL) have emerged as the key recipe for transforming large language models (LLMs) into robust reasoners. We argue that this combined success is driven by **compositional generalization**, which we formalize through a **hierarchical latent selection model**. In this framework, reasoning traces are generated by a cascade of discrete latent selection variables corresponding to reusable atomic modules, including both skills (local operations) and routing mechanisms (how intermediate information is selected, reused, and composed). Within this model, we theoretically show that SFT and RL play asymmetric, complementary roles: SFT supplies the raw module materials in compositional traces, and RL decomposes those traces to identify the latent atomic modules and enable compositional generalization. We design controlled experiments to validate this theory. Our results demonstrate that RL can extract atomic modules from compound traces supplied by SFT and recombine them to solve new configurations. Moreover, we find that training on compound traces yields stronger generalization than training on isolated atomic modules. Finally, we investigate the relationship between SFT and RL data and identify an effective protocol in which SFT ensures coverage of all atomic modules through compositional traces, while RL focuses on novel compositions outside the SFT support to drive exploration.

---
[*]Equal contribution   [1] Carnegie Mellon University   [2] Mohamed bin Zayed University of Artificial Intelligence   [3] Institute of Foundation Models   [4] University of Michigan. Correspondence to: Lingjing Kong <lingjink@cs.cmu.edu>.

*Proceedings of the $43^{rd}$ International Conference on Machine Learning*, Seoul, South Korea. PMLR 306, 2026. Copyright 2026 by the author(s).

## 1  Introduction

Post-training pipelines that pair supervised fine-tuning (SFT) with reinforcement learning (RL) are widely credited with the recent step change in reasoning performance of modern large language models (LLMs) (OpenAI, 2024; Guo et al., 2025; DeepMind, 2025). Empirically, SFT alone often imitates a small set of canonical "golden" traces and degrades when familiar reasoning steps must be recombined in unfamiliar ways, whereas SFT followed by RL handles such out-of-distribution (OOD) compositions far more reliably (Zhang et al., 2025b). However, the mechanism underlying this combined success remains unclear: what does each stage contribute, and what data should each be paired with? We address both questions through a latent-variable account of compositional reasoning.

Prior work has primarily approached the generalization behavior of RL through empirical evaluation. One line of work tracks improvements from RL using standard reasoning benchmarks (e.g., *pass@k* on math and code) (Yue et al., 2025; Wen et al., 2025; Yeo et al., 2025). Another line constructs controlled synthetic reasoning tasks, spanning mathematics (Zhang et al., 2025b), algorithmic coding (Sun et al., 2025a), string transformation (Yuan et al., 2025), and spurious-reward settings (Shao et al., 2025), to isolate behavioral effects under known ground-truth structure. (See Appendices A and B for additional related work and detailed comparisons.) While these studies establish that RL improves OOD accuracy, they leave open *how* RL reshapes reasoning traces to enable *compositional* generalization.

We propose a latent-variable explanation. We model a reasoning trace as the output of a **hierarchical latent selection model** (§2), in which a problem descriptor, the abstract specification of the task encoded in its problem statement, induces a hierarchy of discrete selection variables that choose reusable atomic modules (Figure 1). We distinguish **skills** (local operations) from **routing mechanisms** (how intermediate results are composed). Under this view, SFT-curated traces *supply the raw module material* but leave the modules statistically entangled, because skills and routing co-occur

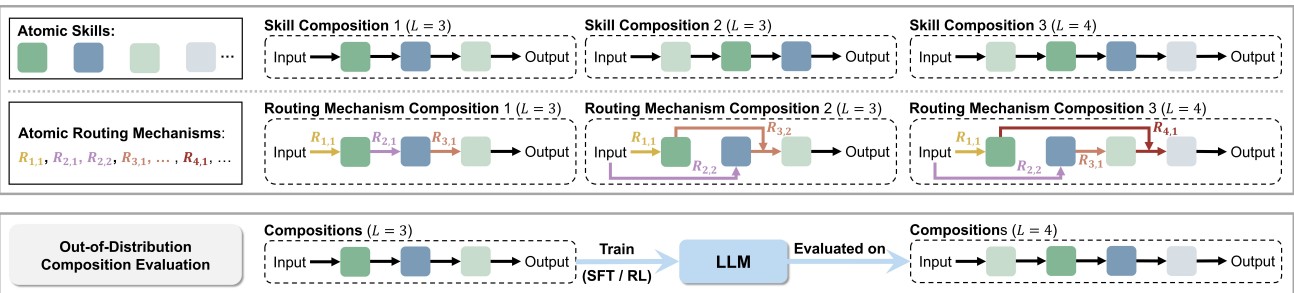

*Figure 1.* **Illustration of atomic skills and routing mechanisms, their compositional traces, and out-of-distribution (OOD) composition evaluation.** A reasoning trace is generated by composing two kinds of reusable atomic modules. (i) *Atomic skills* (colored squares; *top-left inventory*) are local operations on intermediate state, e.g., applying a rewrite rule, performing an arithmetic step, or executing a string transformation. (ii) *Atomic routing mechanisms* (colored arrows, e.g., $R_{1,1}$, $R_{2,i}$, $R_{3,j}$, $R_{4,k}$, ...; *middle-left inventory*) determine how intermediate information is selected and forwarded between skills (e.g., consume the previous output, reuse an earlier intermediate, skip, or branch); the subscript $R_{i,j}$ indexes a routing module by its position $i$ in the chain and an option label $j$. A *composition of length $L$* is a configuration that fills $L$ skill slots and specifies the routing pattern that wires them. **Top row.** Three *skill compositions* (depths $L = 3, 3, 4$) vary which atomic skills occupy the slots while holding the routing template fixed, exhibiting skill recombination. **Middle row.** Three *routing-mechanism compositions* (depths $L = 3, 3, 4$) reuse the same skill inventory while varying the arrow pattern that wires them, exhibiting routing as a separate compositional axis. **Bottom row.** Our OOD composition evaluation: a model is post-trained (SFT and/or RL) on compositions at $L_{\text{train}} = 3$ and evaluated on longer or novel compositions at $L = 4$ whose skill–routing combinations are absent from training. The figure motivates the central message: compositional generalization requires post-training to identify skills and routing mechanisms as *separate* reusable atoms so they can recombine on unseen problem descriptors.

almost deterministically in canonical demonstrations. RL generates trajectory variation under reward that *decomposes* those compound traces into reusable atomic modules.

In §3 we prove that, under mild conditions, the latent atomic modules and their dependency structure are identifiable from the observable distribution over traces (Theorem 3.1). Once identified, these modules recombine on novel compositions whose required interfaces have been locally witnessed in training (Theorem 3.4).

Guided by the theory, we run controlled interventions on the synthetic string transformation tasks following (Yuan et al., 2025), where we use transformation functions as atomic skills and define routing mechanisms as input structure (§4). We find that RL discovers *atomic modules* from *compound traces*, recombining them to solve novel tasks. Specifically, RL on compound traces matches the atomic-task accuracy of direct atomic supervision while also generalizing to unseen compositions (§4.2 and §4.3). We also find that composability must be learned from *composed* experience: withholding an atom (skill or router) from RL can only be repaired by re-injecting compositions that use it, not by isolated atom-only data (§4.4). Finally, the strongest OOD performance occurs when SFT covers the atomic inventory while RL focuses on novel compositions *beyond* the SFT support, with minimal SFT–RL overlap (§4.5). The contributions of this work are:

1. **Hierarchical Latent Selection Model of reasoning.** We introduce a latent-variable model that separates atomic *skills* from *routing mechanisms* and frames post-training as latent structure identification that enables more flexible reuse and recombination.

2. **Theory of identification and composition.** We give

sufficient conditions under which the latent selection hierarchy is provably identifiable from observed traces (Theorem 3.1) and show how identified local compatibility relations compose to guarantee modular compositional generalization via local witnesses (Theorem 3.4). A support analysis (Propositions 3.2 and 3.3) helps clarify the SFT/RL division of labor.

3. **Controlled evidence and data-design implications.** Through systematic interventions on synthetic string-transformation tasks, we verify that RL decomposes compound traces supplied by SFT into reusable atoms, characterize when compound traces and trajectory diversity matter most, and derive practical guidance for designing SFT/RL curricula: SFT ensures coverage of all atomic modules through compositional traces, while RL focuses on novel compositions outside the SFT support (§4.5). Open-source SFT/RL model pairs show the same signature, with increased recombination of abstracted step-level skills after RL (§4.6).

## 2 Reasoning as a Hierarchical Latent Selection Model

We model reasoning as a structured generation process that repeatedly *selects* reusable atomic modules. Intuitively, solving a problem typically involves: (i) choosing a high-level plan, (ii) deciding *what intermediate information to use next* (e.g., reuse the previous result, recall a definition, branch on a condition), and (iii) applying a local operation (e.g., add, substitute, compare). We capture these choices with a **hierarchical latent selection model**, in which discrete latent variables select atomic modules at multiple levels and

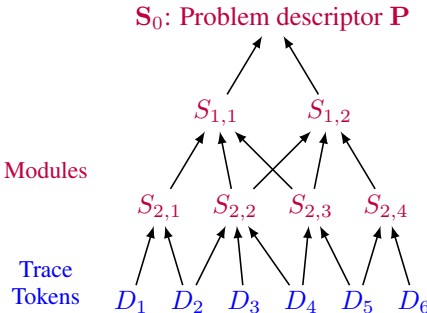

$S_0$: Problem descriptor $\mathbf{P}$

Modules

Trace Tokens

*Figure 2.* **Hierarchical Latent Selection Model.** A problem descriptor $\mathbf{P}$ induces a hierarchy of discrete latent selection variables $\mathbf{S}$ which generate an observable reasoning trace $\mathbf{D}$. In §3 we study when this latent structure can be *identified* from observed $(\mathbf{P}, \mathbf{D})$. Appendix K gives a concrete algebra example illustrating skills, routing mechanisms, and identification conditions.

ultimately generate an observable reasoning trace.

## 2.1 The Hierarchical Latent Selection Model

**Observed problem descriptor $\mathbf{P}$.** Each instance comes with a natural-language problem statement. For analysis, we view this statement as specifying an *abstract descriptor* $\mathbf{P}$ that encodes the highest-level constraints of the task (e.g., what must be computed or proven, which domain constraints apply). We condition on this descriptor throughout.

**Latent selection variables $\mathbf{S}$.** Below $\mathbf{P}$ lies a hierarchy of discrete latent variables $\mathbf{S} = [\mathbf{S}_1, \ldots, \mathbf{S}_L]$. A realization of $\mathbf{S}$ specifies *which atomic modules are used* and *how they are composed*. For example, for a symbolic manipulation task, a high-level selection might correspond to choosing a solution strategy (e.g., substitution vs. factoring), while deeper selections correspond to choosing concrete steps (e.g., expand, simplify, isolate a variable).

**Observable trace $\mathbf{D}$.** The bottom-level variables are the observed tokens $\mathbf{D} = [D_1, \ldots, D_T]$ in the reasoning trace, generated by higher-level selection modules. The arrows in Figure 2 are drawn in the identification direction, from lower-level variables toward higher-level summaries, while sampling proceeds top-down from $\mathbf{P}$ through $\mathbf{S}$ to $\mathbf{D}$.

**Atomic modules: skills and routing mechanisms.** The latent selections $\mathbf{S}$ index a library of reusable atomic modules. We will use the following two terms throughout:

- **Skills (local operations):** modules that perform a local transformation (e.g., add/subtract, apply a rewrite rule, compute a derivative, compare two quantities).
- **Routing mechanisms:** modules that determine how intermediate information is selected, reused, and composed (e.g., carry the previous intermediate result forward, retrieve prior assumptions, choose which subexpression to operate on next, branch/loop patterns).

For instance, in a multi-step algebra problem, a routing mechanism might select "reuse the isolated expression for $x$" as the next input, while a skill applies "substitution."

*Remark* 2.1 (Why "selection"?). Reasoning traces must satisfy coupled constraints across steps. A purely top-down causal story often needs many within-level dependencies to enforce such coherence. In the selection reading (Figure 2), higher-level variables act as discrete summaries of their children: a parent state exists *because* its lower-level neighborhood forms a globally consistent configuration. Our identification proofs use this bottom-up view (Appendix C).

## 2.2 Compositional Generalization and Latent Structure Identification

We study conditional generation of reasoning traces from descriptors. Let $p^\star(\mathbf{D} \mid \mathbf{P})$ be the ground-truth distribution over valid traces, and $\hat{p}(\mathbf{D} \mid \mathbf{P})$ be a learned estimate.

**Training support and compositions.** Let $p_{\text{train}}(\mathbf{P})$ be the training distribution over descriptors, and define its support

$$\Omega_{\text{supp}} := \text{supp}\left(p_{\text{train}}(\mathbf{P})\right) \subseteq \Omega, \tag{1}$$

where $\Omega$ denotes the descriptor space. In many settings, the training support $\Omega_{\text{supp}}$ contains descriptors that activate certain constraints in isolation but not all combinations. As a simple example, if $\mathbf{P} = (P_1, P_2)$ indicates whether two requirements are present, one might observe $(1, 0)$ and $(0, 1)$ during training but never $(1, 1)$ in composition.

A canonical "closure" of $\Omega_{\text{supp}}$ is the Cartesian-product:

$$\begin{aligned} \Omega_{\text{CP}} &:= [\Omega_{\text{supp}}]_1 \times \cdots \times [\Omega_{\text{supp}}]_K, \\ [\Omega_{\text{supp}}]_k &:= \{p_k : (p_1, \ldots, p_K) \in \Omega_{\text{supp}}\}, \end{aligned} \tag{2}$$

which contains all coordinate-wise recombinations of attribute values seen in training. More generally, we will consider a *compositional space* $\Omega_{\text{comp}}$ satisfying

$$\Omega_{\text{supp}} \subsetneq \Omega_{\text{comp}} \subseteq \Omega_{\text{CP}}, \tag{3}$$

where $\Omega_{\text{comp}}$ can exclude incompatible combinations. Under this formulation, compositional generalization refers to the following extrapolation problem.

**Definition 2.2** (Compositional Generalization). Suppose a model $\hat{p}(\mathbf{D} \mid \mathbf{P})$ matches the true conditional distribution on the training support $\Omega_{\text{supp}}$, i.e.,

$$\hat{p}(\mathbf{D} \mid \mathbf{P}) = p^\star(\mathbf{D} \mid \mathbf{P}), \qquad \forall \mathbf{P} \in \Omega_{\text{supp}}. \tag{4}$$

We say the model achieves *compositional generalization* over $\Omega_{\text{comp}}$ if it also matches the true conditional distribution on a strictly larger set $\Omega_{\text{comp}} \supset \Omega_{\text{supp}}$. We call a descriptor $\mathbf{P}$ *composable* if $\mathbf{P} \in \Omega_{\text{comp}}$.

**Latent structure identification.** The descriptor-to-trace map $p^\star(\mathbf{D} \mid \mathbf{P})$ is mediated by the latent hierarchy $\mathbf{S}$ in Figure 2. Since $\mathbf{S}$ is unobserved, learning compositional generalization hinges on whether training can recover (up to an appropriate equivalence) a representation that behaves like the underlying latent modules. We therefore study **latent structure identification**: when and how the hierarchy of selection variables and their dependency structure can be inferred from observed $(\mathbf{P}, \mathbf{D})$. §3 provides conditions under which the hierarchical selection structure is identifiable and implies compositional generalization over $\Omega_{\mathrm{comp}}$.

# 3 Theoretical Framework: Identification $\rightarrow$ Composition

**From the model to the mechanism.** The hierarchical latent selection model in §2 turns compositional generalization into two concrete questions. First, **identification**: can training recover reusable modules, namely skills and routing mechanisms, from observed $(\mathbf{P}, \mathbf{D})$ pairs? Second, **composition**: once those modules are recovered, when can their locally learned interfaces recombine correctly on compositionally novel descriptors $\mathbf{P} \in \Omega_{\mathrm{comp}} \setminus \Omega_{\mathrm{supp}}$?

## 3.1 Identification: When can we recover the latent module hierarchy?

**What is being identified.** Each latent node in Figure 2 corresponds to selecting an atomic module, either a *skill* (local operation) or a *routing mechanism* (how intermediate information is selected and reused). Identification asks whether $P(\mathbf{D} \mid \mathbf{P})$ pins down (i) the latent nodes, (ii) their adjacency, and (iii) the local conditionals up to componentwise relabeling. An identified module is reusable wherever its local prerequisites are met, whereas an entangled trace template remains tied to its training context.

**Theorem 3.1** (Informal: Identification of latent reasoning modules). *Consider the hierarchical selection model in Figure 2. Under Condition C.1 (Appendix C), the latent selection variables $\mathbf{S}$ and their adjacency structure are identifiable from the observed distribution $P(\mathbf{D} \mid \mathbf{P})$, up to component-wise relabeling of discrete states.*

**From identification to two post-training algorithms.** Theorem 3.1 gives sufficient conditions for identification. We focus on local events: node assignments and their child contexts that make the assignment meaningful,

$$\mathcal{E} = \{U = u, \mathrm{Ch}(U) = v_{\mathrm{Ch}(U)}\}. \tag{5}$$

For a node $U$ and a tuple of children values $v_{\mathrm{Ch}(U)}$, define

---

**Assumption Intuition for Condition C.1**

- **Observable signatures** (Condition C.1-vi). Different module states have different observable predictive signatures, so local choices force different downstream effects. *Example:* From $2(x + 3) = 14$, the next line $x + 3 = 7$ ("divide first") vs. $2x + 6 = 14$ ("expand first") is a tiny local cue that reveals the chosen step and forces different follow-ups.

- **Restricted choice sets** (Condition C.1-iii). Valid reasoning occupies a structured subset of the combinatorial space: given its context, a module has only a small set of plausible next choices. *Example:* In $58 + 67$, after computing $8 + 7 = 15$, the "write digit / carry" choice is essentially forced to

  (write 5, carry 1). All other (digit, carry) pairs are invalid.

- **Neighborhood coverage** (Condition C.1-v). Each module must have a pure local cue and enough observed comparison context to decouple it from its typical co-occurrences. *Example:* If `sort()` only ever appears right before "take the median," it can be mistaken as one fused routine; seeing `sort()` also before "binary search" and "deduplicate" provides distinct neighborhoods that isolate sorting as a reusable module.

the *training-witnessed local compatibility set*

$$\mathcal{W}_U\big(v_{\mathrm{Ch}(U)}\big) := \Big\{ u \ : \ \exists p \in \Omega_{\mathrm{supp}} \text{ s.t.}$$

$$p^\star(U\!=\!u,\ \mathrm{Ch}(U)\!=\!v_{\mathrm{Ch}(U)} \mid \mathbf{P}\!=\!p) > 0 \Big\}. \tag{6}$$

That is, $u \in \mathcal{W}_U(v_{\mathrm{Ch}(U)})$ records that $\mathcal{E}$ is observed under some compatible training descriptor.

For example, consider solving a system of equations. After deriving $x = 2y + 1$, the trace may need to reuse this expression as an intermediate and substitute it into the second equation. Here $\mathcal{E}$ combines a skill, substitution, with a routing decision, sending an earlier intermediate result to the later equation where it is needed. In latent-selection notation, $U$ can represent the local choice that routes the derived expression forward, while $v_{\mathrm{Ch}(U)}$ represents the surrounding context that makes the substitution valid.

**What SFT traces provide, and what they leave entangled.** Supervised fine-tuning *supplies the raw module material*: each compositional trace exhibits the atomic skills and routing mechanisms in working combination, the substrate the model needs to ever reach those modules during rollouts. However, when each prompt comes with a single canonical chain-of-thought, the atomic modules co-occur almost deterministically with their usual contexts, and a learner can fit $p^\star(\mathbf{D} \mid \mathbf{P})$ on $\Omega_{\mathrm{supp}}$ while never explicitly separating *which module* produced *which part* of the trace. The supervised distribution therefore gives the *materials* but not the *decomposition*: module identities remain entangled with their typical neighborhoods because the local events that would separate them lie outside the SFT trace support—a *hidden-support obstruction* formalized in Proposition 3.2.

**Informal Proposition 3.2** (SFT hidden support). *Fix a prompt marginal $\mu^\star$ and the true prompt-trace law $Q^\star(p, d) = \mu^\star(p)p^\star(d \mid p)$. For each prompt $p$, suppose SFT reveals only a subset $A_S(p)$ of valid traces, with mass*

$$s(p) := \sum_d p^\star(d \mid p)\mathbf{1}\{d \in A_S(p)\}. \qquad (7)$$

*Let $q_S(d \mid p)$ be the conditional law obtained by renormalizing $p^\star(d \mid p)$ on $A_S(p)$, and let $\widetilde{Q}(p, d) = \mu^\star(p)q_S(d \mid p)$. Then $Q^\star$ and $\widetilde{Q}$ induce identical SFT observations, yet their trace laws differ by the hidden trace mass:*

$$\|Q^\star - \widetilde{Q}\|_1 = 2\,\mathbb{E}_{\mu^\star}[1 - s(p)]. \qquad (8)$$

*Any identification-critical local event that occurs only outside $A_S(p)$ is absent under $\widetilde{Q}$ and therefore cannot be certified from SFT observations alone.*

In the language above, $A_S(p)$ is the SFT-revealed neighborhood of $p$; events outside $A_S(p)$ are the local cues that would disentangle a module from its typical context, and the proposition says SFT alone cannot rule them in or out.

**How RL decomposes traces by enriching identification-critical local events.** RL re-samples rollouts under the verifier and reweights them by reward, so when an SFT-hidden event is *reachable* by the current policy and traces containing it succeed more often than nearby variants that omit it, reward-positive rollouts *enrich* the event, exposing a local difference that SFT alone could not certify. This is the support-expansion mechanism by which RL decomposes compound traces into identifiable modules.

**Informal Proposition 3.3** (RL enrichment of useful local events). *Fix $\mathbf{P} = p$ and $\mathcal{E} = \{U = u, \mathrm{Ch}(U) = v_{\mathrm{Ch}(U)}\}$, let $\hat{p}_0(\mathbf{D} \mid \mathbf{P})$ denote the current model before the RL update, and define the reward gap $\Delta_\mathcal{E}(p) := \Pr(R{=}1 \mid \mathcal{E}, \mathbf{P}{=}p) - \Pr(R{=}1 \mid \mathcal{E}^c, \mathbf{P}{=}p)$. Suppose:*

1. ***Rollout reachability.*** $0 < \hat{p}_0(\mathcal{E} \mid \mathbf{P} = p) < 1$: *the current policy visits $\mathcal{E}$ with positive probability.*

2. ***Reward informativeness.*** $\Delta_\mathcal{E}(p) > 0$ and $\Pr(R{=}1 \mid \mathbf{P}{=}p) > 0$: *traces containing $\mathcal{E}$ succeed more often than traces that omit it, with non-degenerate base success.*

*Then reward-positive rollouts enrich $\mathcal{E}$,*

$$\Pr(\mathcal{E} \mid R = 1, \mathbf{P} = p) - \hat{p}_0(\mathcal{E} \mid \mathbf{P} = p) > 0, \qquad (9)$$

*with magnitude proportional to $\hat{p}_0(\mathcal{E} \mid \mathbf{P}{=}p)\big(1 - \hat{p}_0(\mathcal{E} \mid \mathbf{P}{=}p)\big)\Delta_\mathcal{E}(p)$, the local policy-gradient signal given descriptor $p$ that upweights $\mathcal{E}$-traces.*

**Interpretation.** Each condition closes off a way the update could be vacuous. Condition (1) (*reachability*) says the current policy already places some, but not all, of its rollout mass on $\mathcal{E}$; if rollouts never reach $\mathcal{E}$ or always do, reward-conditioned resampling has no mass to redistribute. Condition (2) (*reward informativeness*) says the verifier actually prefers $\mathcal{E}$-containing traces; if rewards are uninformative about $\mathcal{E}$, reweighting cannot favor it. Under both, conditioning on $R{=}1$ strictly increases the rate at which $\mathcal{E}$ appears, which is what we mean by RL decomposing the entangled trace into a locally identifiable choice.

**Division of labor.** Proposition 3.2 and Proposition 3.3 together describe a *division of labor*: SFT supplies the raw module material but cannot certify events outside its trace support, while RL targets exactly those SFT-hidden events whenever they are rollout-reachable and reward-informative. The empirical question is therefore *what data each algorithm is paired with*, the prescription tested in §4.5; Appendix D gives the complete formal statements and proofs.

### 3.2 Composition: When do identified modules recombine out of distribution?

Identification alone does not guarantee compositional generalization: a novel prompt can force familiar modules to meet at *new interfaces*. Reading Figure 2 bottom-up, a higher-level selection restricts which lower-level values are compatible; when several children constrain the same parent, it must simultaneously satisfy *all* child-imposed constraints.

**Theorem 3.4** (Informal: Compositional generalization via local witnesses). *Assume the latent model in Figure 2. If every configuration of latent parents arising under $p_{\mathrm{new}} \in \Omega_{\mathrm{comp}}$ admits a local witness, i.e., for every induced local family $(U = u, \mathrm{Ch}(U) = v_{\mathrm{Ch}(U)})$ under $p_{\mathrm{new}}$,*

$$u \in \mathcal{W}_U\big(v_{\mathrm{Ch}(U)}\big), \qquad (10)$$

*then the locally learned constraints can indeed be composed consistently without contradiction under $p_{\mathrm{new}}$.*

**Interpretation.** Condition (10) requires only that whenever a novel prompt makes several higher-level variables $\mathrm{Ch}(U)$ meet at a shared parent $U$, training has witnessed a compatible value $u \in \mathcal{W}_U(v_{\mathrm{Ch}(U)})$. This condition refers back to the support mechanism above. Proposition 3.2 says that missing local events cannot be certified from censored supervised traces alone, while Proposition 3.3 says that reachable and reward-informative events are enriched by RL. This precisely explains why compound traces are useful in practice when they exercise module interfaces, and RL gains concentrate on off-support compositions when rollouts expand the local witness sets needed by Theorem 3.4. If two child modules have never shared a parent value during training, the witness set is empty and (10) fails.

**Connection to our empirical protocol.** §4 empirically tests these messages by varying compound-only training, held-out skills, held-out routing mechanisms, and the overlap

between supervised and RL support:

- **Decomposition into atoms.** §4.2 trains on compound-only traces and evaluates both atomic recovery and challenging unseen-composition transfer.

- **Material vs. decomposition.** Sections 4.3 and 4.4 withhold atomic skills and routing mechanisms, showing that recovery requires re-injecting *compositional* traces; pure atomic re-injection is insufficient at depth.

- **Algorithm–data pairing.** §4.5 varies the SFT–RL composition-set relationship, finding that disjoint SFT/RL data dominate on unseen compositions.

## 4 Empirical Findings: SFT-Supplied Materials, RL Decomposition, and the Data Pairing That Connects Them

To validate our theoretical analysis, we run controlled experiments on synthetic string transformation tasks. By precisely intervening in the data structure of training samples during SFT and RL, we systematically control the availability of atomic modules and their compositions, enabling a direct examination of how SFT-supplied compositional traces and RL together support compositional generalization, and of which SFT–RL data pairing is most effective.

### 4.1 Experimental Setup

**Tasks.** Our experimental task follows the synthetic string transformation introduced in prior work (Yuan et al., 2025), in which atomic modules and their compositions are precisely defined. Specifically, the task consists of a fixed set of deterministic string transformation functions $f_i(x)$, where $i$ indexes the functions, which serve as atomic skills. In addition, we define function input structures as atomic routing mechanisms. For example, given a composition of three steps, $g_1^3(y_1, y_2) = y_1$ and $g_2^3(y_1, y_2) = y_1 + y_2$ denote two distinct routing mechanisms. The former sends only the first-step output to the third step, while the latter sends both the first-step and second-step outputs. Here, $y_1$ and $y_2$ are the outputs of steps 1 and 2, respectively. Compositional reasoning instances are constructed by nesting these functions and routing structures to varying depths (e.g., $f_7(f_{11}(x), f_3(x))$). We report performance as the accuracy of the predicted string after applying the specified transformation. We use the 24 atomic skills from (Yuan et al., 2025) and design 10 new atomic routing mechanisms. The compositional depth, referred to as level $L$, directly controls task difficulty and enables systematic evaluation of generalization to unseen compositions.

**Training and Evaluation.** The training consists of two stages over the same prompt family. In the first stage, SFT trains the model on correct reasoning traces. In the second

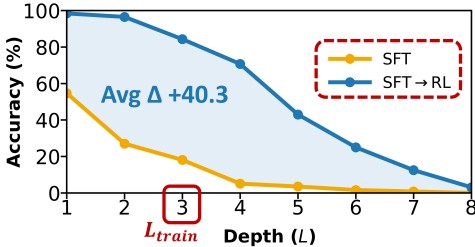

*Figure 3.* **Accuracies of SFT and SFT+RL across compositional depths.** Models are trained on traces with $L_{\text{train}} = 3$. RL substantially improves test-time compositional generalization performance over different compositional depths.

stage, RL samples rollouts for the same task distribution and receives reward only from the final output string. To construct the chain-of-thought (CoT) training data for SFT, we generate ten reasoning-inclusive responses per problem and retain only those with correct final answers as the training data. Both SFT and RL may train the model using either atomic modules (skills and routing) or compositional reasoning traces based on atoms, where we modify the data structure to analyze the underlying learning mechanisms. For the OOD evaluation setting, generalizability is assessed by testing models on compositional instances whose combinations are not observed during training in either the SFT or RL stages. In what follows, unless explicitly noted, "compositions" in evaluation denote unseen compositions. Appendix F gives the full model, data, training, and hyperparameter details for all experiments.

**Why synthetic tasks.** Synthetic tasks are not a substitute for real benchmarks; they are a tool for isolating causal mechanisms. They let us manipulate support coverage and composition structure without confounds, directly testing whether RL produces modules or memorizes traces; §4.6 complements this with open-source SFT/RL analyses.

### 4.2 Finding 1: RL Decomposes Traces into Atoms and Recombines Them for Generalization

**Task setting.** To examine whether RL can decompose compound reasoning traces into reusable atomic skills, we restrict both the SFT and RL stages to observe only compound reasoning traces at a fixed depth ($L = 3$), such as $f_3(f_2(f_1(x)))$. For evaluation, we first assess the performance of models trained with and without RL on individual atomic functions (e.g., $f_1, f_2, f_3$) (Figure 3 at $L = 1$). We then evaluate generalization to unseen compositions at increasing trace depths ranging from $L = 2$ to $L = 8$ (Figure 3). In addition, we report performance on both seen and unseen compositions for comparison (Figure 4).

**Observations and discussions.** The accuracy at $L = 1$ in Figure 3 shows that models perform well on atomic skills with only $L_{\text{train}} = 3$ training traces, where RL adds a 43.7% gain over SFT-only training. On unseen compositions, SFT

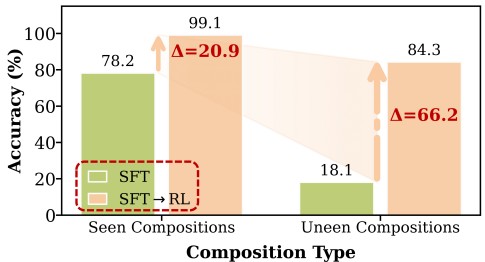

*Figure 4.* **Comparison of model accuracy on seen and unseen compositions.** RL provides limited gains on seen compositions while delivering substantially larger gains on unseen compositions.

drops rapidly with depth while RL remains higher accuracy across depths (avg $+40.3\%$); the SFT-vs-RL gap is much larger on unseen than seen compositions (Figure 4).

*Table 1.* **Effect of RL data structure on OOD compound traces.** Starting from the same SFT model trained on $L = 3$ traces, we run RL for 300 steps with either atomic modules ($L = 1$) or compound traces ($L = 3$), then evaluate on OOD compound traces ($L = 4$).

| Training setting | Accuracy (%) | Gain |
|---|---|---|
| SFT baseline | 4.8 | – |
| SFT+RL, atomic modules | 14.8 | +10.0 |
| SFT+RL, compound traces | 42.6 | +37.8 |
| Compound advantage | – | +27.8 |

Table 1 isolates the role of compound RL traces under a fixed SFT initialization and RL budget. Atomic-module RL improves over SFT by reinforcing reusable operations, but compound-trace RL gives a much larger gain because it exposes the local interfaces between modules, matching the witness-based composition view in Theorem 3.4.

> **Takeaway 1**
>
> RL decomposes SFT-supplied compound traces into reusable atoms and recombines them for unseen compositions; under the same SFT initialization, compound-trace RL yields much larger OOD gains than atomic-only RL.

### 4.3 Finding 2: Composability Requires Combinational Exposure during RL

**Task setting.** To examine how RL performs when atomic knowledge is partially absent, we start from the base setting where SFT and RL both train on $L = 3$ compositional traces covering all atomic skills (defined transformation functions), then remove 8 atomic skills (1/3) and exclude all compositions containing them from RL. We compare three RL settings: no re-injection, re-injection of isolated atomic skills, and re-injection of compositional traces of varying depth. Model performance is evaluated on compositional traces containing the removed skills across $L = 1$ to $L = 4$.

**Observations and discussions.** As shown in Figure 5, when a subset of atomic skills is removed from RL training, models trained without any re-injection of the corresponding

compositions exhibit consistently lower accuracy across evaluation depths. In contrast, re-injecting compositional traces that contain the held-out atomic skills leads to substantial performance recovery. These comparisons suggest that the atomic knowledge is foundational for effective RL training. When certain atomic skills are absent, RL alone is insufficient to recover them in isolation. One notable observation is that directly re-injecting atomic skills does not generalize well to traces with greater compositional depth. This is because exposure to isolated atomic skills alone does not teach the model how to recombine them to solve new compositions, whereas re-injecting compositional traces provides combinational exposure during RL that supports deeper generalization. Appendix J reports 1000-step convergence runs for this experiment.

> **Takeaway 2**
>
> Atomic knowledge is necessary but not sufficient: models must encounter atoms within compositional contexts to learn to recombine them and generalize systematically to deeper, unseen compositions.

### 4.4 Finding 3: A Shared Learning Mechanism for Skills and Routing

We repeat the experiment for routing mechanisms, which determine which intermediate outputs are reused.

**Task setting.** Beyond atomic skills, we further examine how atomic routing mechanisms are learned. Following the task setting in Section 4.3, we remove one atomic routing mechanism during RL training. Specifically, we exclude RL training compositions that contain the held-out routing mechanism, then re-inject lower-depth compositions involving that mechanism and evaluate on unseen compositions of depths $L = 3$ and $L = 4$ that exercise the held-out routing.

**Observations and discussions.** Figure 6 shows a pattern similar to the skill-based experiments: removing atomic routing mechanisms during RL training degrades generalization, while re-injecting compositional traces that contain the held-out mechanisms largely restores performance. Interestingly, re-injecting routing compositions with lower depth leads to stronger improvements than deeper compositions. This suggests that simpler routing compositions make the underlying atomic routing mechanisms easier to recover.

### 4.5 Finding 4: Pairing SFT and RL Data: The Distribution Relationship That Matters

Findings 2 and 3 show that skills and routers are reusable only in the right compositional contexts. We now ask how the SFT and RL composition sets should relate so that SFT covers the atomic inventory while RL explores beyond it.

**Experiment I.** We explore how the relationship between the

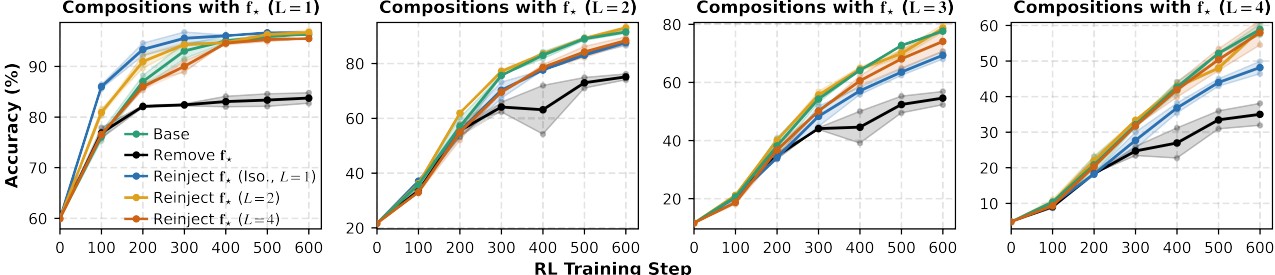

*Figure 5.* **Accuracy on compositional traces involving held-out atomic skills across varying trace depths and re-injection strategies.** The solid line represents the average result over two independent runs with different random seeds, while the shaded region shows the empirical variation across runs. "Iso." denotes "Isolated". All settings share the same SFT model trained on $L = 3$ traces. We remove compositions involving the held-out skill $f_\star$ from the base RL corpus, then augment with re-injection traces, either isolated $f_\star$ at $L = 1$ or compositions containing $f_\star$ at varying depths. RL proceeds from the SFT checkpoint on this modified corpus; no second finetuning round. Atomic knowledge is essential for effective RL training, and removed atomic knowledge can be re-injected only through compositions that involve the corresponding atoms, rather than through isolated atomic skills.

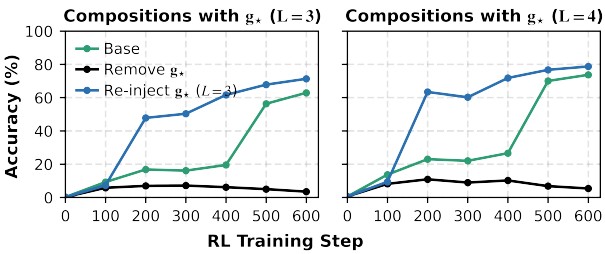

*Figure 6.* **Accuracy comparison across settings with full routing compositions, atomic routing mechanism removal, and routing mechanism re-injection with lower depth.** We report the performance on unseen compositions with both $L = 3$ and $L = 4$.

SFT and RL data distributions (composition sets) influences atom discovery and reasoning. Specifically, we consider four canonical relationships between the two composition sets, denoted by $\mathcal{C}_{\mathrm{SFT}}$ and $\mathcal{C}_{\mathrm{RL}}$: (1) $\mathcal{C}_{\mathrm{SFT}} \supset \mathcal{C}_{\mathrm{RL}}$, where RL compositions are fully covered by SFT; (2) $\mathcal{C}_{\mathrm{SFT}} \subset \mathcal{C}_{\mathrm{RL}}$, where RL expands beyond the SFT composition set; (3) $\mathcal{C}_{\mathrm{SFT}} \not\subset \mathcal{C}_{\mathrm{RL}}$, $\mathcal{C}_{\mathrm{SFT}} \not\supset \mathcal{C}_{\mathrm{RL}}$ and $\mathcal{C}_{\mathrm{SFT}} \cap \mathcal{C}_{\mathrm{RL}} \neq \emptyset$, where the two composition sets partially overlap; (4) $\mathcal{C}_{\mathrm{SFT}} \cap \mathcal{C}_{\mathrm{RL}} = \emptyset$, where the SFT and RL distributions are disjoint.

We train models with composition depth $L = 3$, and evaluate them on seen compositions, unseen compositions, and a depth extrapolation setting with greater composition depth.

**Observations and discussions.** As shown in Figure 7, different SFT-RL distribution relationships have negligible impact on performance in seen settings, but substantially affect generalization to unseen settings. In particular, the setting $\mathcal{C}_{\mathrm{SFT}} \supset \mathcal{C}_{\mathrm{RL}}$, where SFT already covers the RL composition set, exhibits the weakest unseen performance. By contrast, the disjoint setting $\mathcal{C}_{\mathrm{SFT}} \cap \mathcal{C}_{\mathrm{RL}} = \emptyset$ achieves the strongest generalization. This suggests that RL benefits from compositions that fall outside the SFT composition

*Table 2.* **Performance comparison with and without atomic absence in the SFT stage across seen and unseen compositional settings.** "2"/"3" denotes the composition depth.

| Method | Stage | Seen-2 | Seen-3 | Unseen-2 | Unseen-3 |
|---|---|---|---|---|---|
| No Absence in SFT | SFT | 0.91 | 0.83 | 0.30 | 0.22 |
| | +RL | 0.99 | 0.98 | 0.67 | 0.55 |
| Absence in SFT | SFT | 0.90 | 0.84 | 0.24 | 0.15 |
| | +RL | 0.99 | 0.96 | 0.52 | 0.36 |

support, as reduced overlap encourages exploration beyond supervised patterns. Appendix J reports longer 1000-step training runs showing the same ordering.

**Experiments II–III: SFT data design.** We further analyze how the SFT data shapes RL gains along two axes. *(II) Atom coverage:* we split $131k$ training compositions ($L = 2, 3$) into two partitions where SFT either covers all atomic skills or excludes 4, while RL always uses the full atom set. Partition choice has little effect in-distribution but markedly hurts OOD performance, especially after RL, whenever atoms are absent from SFT (Table 2). *(III) Compositional depth:* varying SFT data from atomic skills ($L = 1$) to compositional traces ($L = 2, 3$), Figure 8 shows that SFT on compositional traces yields stronger OOD generalization, including on atomic-skill tasks ($L = 1$).

> **Takeaway 3**
>
> SFT should cover all atomic modules through compositional traces, while RL should target genuinely novel compositions outside the SFT support.

### 4.6 Complementary Evidence on Open-Source Models

As a complement to the controlled experiments, we test whether the same signature appears in real post-trained LLMs. We collect 643 math problems from MATH-500, AIME 2024/2025, AMC, and GSM8K, generate traces from Qwen3-4B and Qwen3-4B-Thinking-2507, and map each

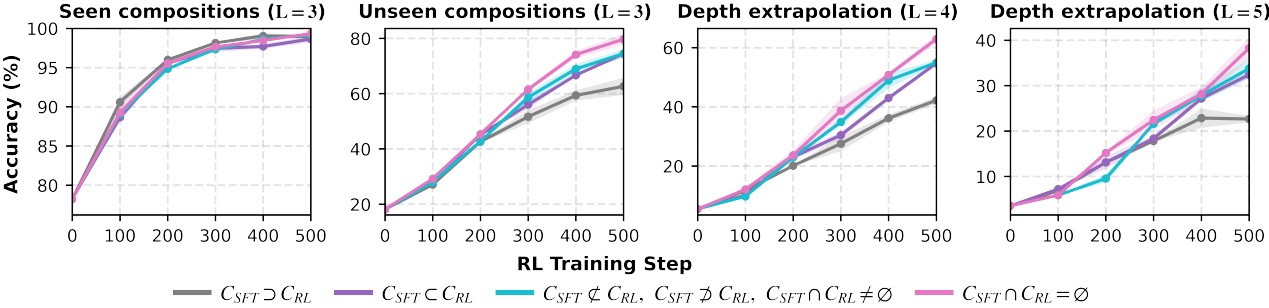

*Figure 7.* **Comparison of models under different SFT-RL data distribution relationships.** We evaluate the performance on seen compositions, unseen compositions, and depth extrapolation to higher compositional levels, respectively. The solid line reports the mean average over two seeds, and the shaded region shows variation across random seeds.

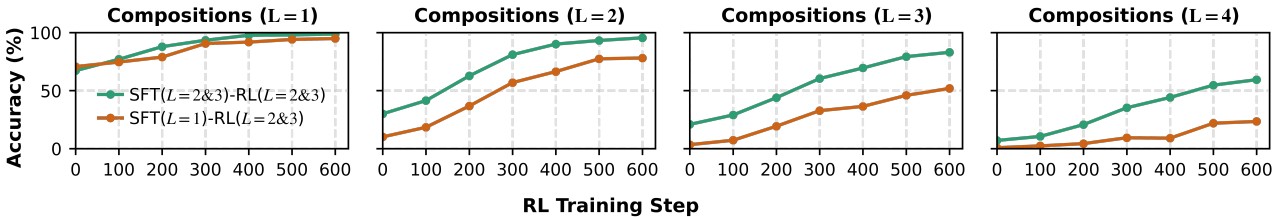

*Figure 8.* **Comparison of SFT training on atomic skills ($L = 1$) versus compositional traces ($L = 2\&3$) across composition depths.** SFT trained on compositional traces exhibits stronger generalization performance.

trace to a sequence of abstract mathematical skills. The RL-tuned model shows richer diversity in short skill $n$-grams ($n = 2, 3$), indicating more flexible local recombination; longer $n$-grams are less diagnostic. Short motifs probe whether a reasoning operation can be paired with different neighboring operations across problems, the real-model analogue of the interface coverage tested above. For example, a pattern that combines "set up equation" with "case split" in one problem and with "substitute back" in another is more suggestive of reusable reasoning pieces than a long repeated template that appears only in one problem family. Appendix G gives the pipeline and full plots.

## 5 Conclusion

We introduce a selection-based view of LLM reasoning, where a prompt induces a latent hierarchy of modular decisions that ultimately produces the observed compositional trace. We showed that SFT and RL play asymmetric, complementary roles: SFT supplies compositional traces that contain the atomic inventory, while RL re-samples those traces under reward and efficiently enriches identification-critical local events the supervised distribution kept entangled, explicitly decomposing the trace into reusable atomic modules with locally compatible interfaces. Our experiments confirm an effective data-pairing protocol: SFT ensures coverage of all atomic modules through compositional traces, while RL focuses on novel compositions outside the SFT support. This division of labor follows from the

identification conditions (Theorem 3.1), the local-witness composition theory (Theorem 3.4), and the support-side conditions for RL enrichment: reward informativeness, rollout reachability, and SFT-hidden trace mass.

**Implications for curriculum design.** The identification–composition picture yields a concrete corollary for practitioners: the SFT and RL stages should be co-designed to cover complementary parts of the composition space. SFT should expose the atomic inventory through compositional traces, since these are the substrate the model later decomposes; RL should target compositions whose interfaces are unseen in SFT, converting reward signal into identification-critical local events. This recasts the SFT–RL distribution as a curriculum hyperparameter, complementing efforts that tune SFT scale and RL learning rate independently.

**Limitations and future work.** Our empirical study is intentionally simplified to enable precise control over compositional structure; extending this coverage-controlled methodology to richer domains (e.g., math, code, and tool use) is an exciting direction. The importance of local interface coverage suggested by our theory also points to future RL objectives and curricula that explicitly target under-explored interfaces, potentially improving sample efficiency and robustness to distribution shift. An open question is whether such an identification-driven curriculum can be discovered adaptively, by tracking which local interfaces the current policy realizes and steering RL exploration toward those where identification gain is greatest.

**Acknowledgments.** We would like to acknowledge the support from NSF Award No. 2229881, AI Institute for Societal Decision Making (AI-SDM), the National Institutes of Health (NIH) under Contract R01HL159805, and grants from Quris AI, Florin Court Capital, MBZUAI-WIS Joint Program, and the Al Deira Causal Education project. We thank Yuke Li for proofreading and the anonymous reviewers for their valuable feedback and suggestions

## Impact Statement

This paper presents work whose goal is to advance the field of Machine Learning. There are many potential societal consequences of our work, none which we feel must be specifically highlighted here.

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

# A Related Work

**Reasoning traces and adaptive computation.** A growing literature studies how large language models generate, control, and benefit from intermediate reasoning. Early methods elicit explicit reasoning traces through scratchpads or auxiliary tokens (Nye et al., 2022; Goyal et al., 2023; Herel & Mikolov, 2024). Subsequent work trains models to insert or suppress such traces using reinforcement learning, enabling conditional computation and adaptive reasoning length (Zelikman et al., 2024; Fang et al., 2025; Zhang et al., 2025c; Xiang et al., 2025). Test-time control over reasoning has been explored through termination criteria and length prediction (Wu et al., 2025; Eisenstadt et al., 2025), selective reasoning for multimodal models (WANG et al., 2025), or external controllers (Wang et al., 2025d). Latent approaches replace explicit chains with continuous or token-based latent variables, often trained via EM-style objectives (Sun et al., 2025b; Phan et al., 2023; Ruan et al., 2025). At the same time, several works question whether the semantic content of reasoning traces is essential: Stechly et al. (2025) show that intermediate tokens need not be meaningful, while Csordás et al. (2025) find that model depth is often underutilized.

**Uncertainty, entropy, and exploration.** Uncertainty has emerged as a key signal for both training and inference-time reasoning. Entropy-aware objectives encourage exploration in RL-based reasoning (Agarwal et al., 2025; Cui et al., 2025; Cheng et al., 2025a; Wan et al., 2025), with evidence that high-entropy or minority tokens disproportionately drive learning dynamics (Wang et al., 2025b;c). This motivates branching or backtracking at uncertain points (Zhu et al., 2025; Zheng et al., 2025; Hou et al., 2025; Lu et al., 2025; Qin et al., 2025), as well as temperature or confidence-based control (Zhu et al., 2024; Li et al., 2025). Complementary work studies calibration and self-verification, predicting correctness from intermediate representations or confidence signals (Zhang et al., 2025a; Yoon et al., 2025; Jang et al., 2025). These methods improve exploration and reliability, but do not address when latent reasoning structure becomes identifiable.

**Interpretable structure and latent representations.** Mechanistic interpretability approaches aim to uncover internal reasoning structure. Sparse autoencoders and related factorization methods extract interpretable features or steering vectors (Wang et al., 2025a; Venhoff et al., 2025), but face challenges from polysemanticity and correlated features (Chanin et al., 2025; Deng et al., 2025). Recent work proposes hierarchical or multi-level sparse architectures and projection-based discovery methods (Balagansky et al., 2025; Muchane et al., 2025; Costa et al., 2025), with emerging identifiability guarantees (Chen et al., 2025b; Cui et al., 2026). These efforts build on classical results in non-

negative matrix factorization and nonnegative rank (Cohen & Rothblum, 1993). Other approaches extract prototypes or latent steering directions by contrasting reasoning and non-reasoning paths (He et al., 2025; Liu et al., 2025b). In contrast, we study how training dynamics themselves enable recovery of latent compositional structure.

**Supervised fine-tuning versus reinforcement learning.** A central question is why reinforcement learning (RL) often generalizes better than supervised fine-tuning (SFT) for reasoning. Empirically, SFT tends to memorize reasoning traces, while RL improves length generalization and transfer (Chu et al., 2025; Huan et al., 2025; Tsilivis et al., 2025). Several works unify or interpolate between SFT and RL objectives (Liu et al., 2025a; Lv et al., 2025; Liu et al., 2025c), analyze their implicit equivalence (Anil et al., 2025; Swamy et al., 2025), or identify coverage as the key determinant of RL success (Chen et al., 2025a; Zhang et al., 2025b). Others show that RL composes atomic skills learned during SFT (Cheng et al., 2025b; Yuan et al., 2025), while distillation-based approaches tend to learn flatter reasoning structures (Xu et al., 2026).

**Positioning.** Most prior work focuses on eliciting longer reasoning traces, improving exploration, or interpreting trained models. In contrast, we formalize reasoning as hierarchical generation with discrete latent selections and show that RL induces coverage over these latent choices. This yields identifiability conditions for recovering compositional structure and explains why RL enables out-of-distribution generalization beyond what supervised objectives alone can achieve.

# B Detailed Discussions with Existing Work

**Comparison with (Yuan et al., 2025).** To clarify the contributions of our work, we compare it with (Yuan et al., 2025). First, leveraging the well-designed atomic skills based on string transformation functions in (Yuan et al., 2025), we adopt the same task framework for controlled experimentation. (It is not our contribution.) Building on these atomic skills, we further introduce explicit routing mechanisms through function input structures, enabling a systematic study of how RL learns routing behavior. Second, while (Yuan et al., 2025) demonstrates that RL enables compositional abilities, we go further by identifying the underlying mechanism through which RL discovers reusable atomic skills. Third, beyond the cross-depth out-of-distribution evaluations considered in (Yuan et al., 2025), we conduct targeted interventions on training compositions to examine (i) how atomic skills and compositional traces influence RL, and (ii) how the relationship between the distributions of SFT and RL affects atomic discovery. Finally, and most importantly, we establish a latent-variable-model-based theoretical framework that formally analyzes the learning mechanism of RL, providing principled insights

that complement and extend prior empirical findings.

**Comparison with (Wen et al., 2025).** (Wen et al., 2025) is another work that also introduces the theoretical analysis for the behavior of RL. Specifically, it provides an *optimization-dynamics* account of RL: under GRPO, a pre-trained model is assumed to satisfy a *logic prior*, so the expected advantage is positive for samples with correct reasoning and negative otherwise, implying RL mainly *reweights* the model toward correct CoTs even when the reward only checks the final answer. In contrast, our theory targets *compositional generalization*: we model reasoning as a *hierarchical latent selection* process (skills + routing) and formalize two technical problems—**identification** (recovering latent modules/structure from $P(\mathbf{D} \mid \mathbf{P})$) and **composition** (recombining learned local constraints on novel descriptors $\mathbf{P} \in \Omega_{\text{comp}} \setminus \Omega_{\text{supp}}$). Accordingly, our main sufficient conditions are (i) identifiability of the latent hierarchy (up to relabeling) and (ii) a "local witness" condition ensuring locally learned constraints can be composed without contradiction on new compositions. This makes RL's theoretical role different: beyond shifting probability mass toward already-valid traces, RL helps by inducing *trajectory diversity* that expands support over module interfaces (creating the local witnesses needed for identification and reliable recombination).

## C   Identification Theory 3.1

### C.1   Formal Theorem 3.1

We study identifiability of the discrete selection variables $\mathbf{S}$ in the hierarchical selection model. Intuitively, each latent selection variable should be a *compressed* representation of its lower-level neighborhood: it should preserve all predictive information about the rest of the trace while using as few states as possible.

**Condition C.1** (Textual Concept Identification Conditions)**.**

i **Faithfulness**. *All (conditional) independence relations in the observed distribution are entailed by d-separation in the underlying graph.*

ii **Rank faithfulness**. *The observed distribution $P$ is rank-faithful to the latent DAG $G$: every nonnegative-rank constraint on a conditional sub-probability table of observed variables that holds in $P$ is entailed by the class of discrete structural models Markov to $G$ with the stated latent cardinalities.*

iii **Restricted choice sets**. *Each selection variable $S_l$ only realizes a strict subset of the combinatorial state space obtained by freely composing its immediate constituents. Concretely,*

$$\text{supp}\, S_l \subsetneq f_{\mathbf{D}\to S_l}\big(\Omega^{n(\text{Pa}(S_l))}\big), \qquad (11)$$

*where $f_{\mathbf{D}\to S_l}$ denotes the (level-l) composition map from lower-level variables to $S_l$.*

iv **Deterministic coarsest selection**. *For each latent node $S$, $S$ is the unique coarsest deterministic statistic of $\text{Pa}(S)$.*

v **Neighborhood coverage**. *For every selection variable $S_l$, there exists a diagnostic pure parent $D^\star \in \text{Pa}(S_l)$ such that $\text{Ch}(D^\star) = \{S_l\}$ and*

$$|\text{supp}\, D^\star| > |\text{supp}\, S_l|. \qquad (12)$$

*Moreover, the observed variables outside this diagnostic cue provide a non-vacuous comparison table. Writing*

$$C_{D^\star} := \text{CoPa}(D^\star), \qquad R_{D^\star} := \mathbf{D} \setminus \big(\{D^\star\} \cup C_{D^\star}\big), \qquad (13)$$

*there exists $c \in \text{supp}(C_{D^\star})$ such that*

$$\big|\text{supp}\,\big(R_{D^\star} \mid C_{D^\star} = c\big)\big| \geq |\text{supp}(S_l)|. \qquad (14)$$

*Thus each latent selection has a pure local cue, and the opposite side of the rank table is large enough for a rank drop to reveal a genuine latent bottleneck.*

vi **Observable module signatures**. *For each selection variable $S$, write $\text{Pa}(S) = \mathbf{D}^S \cup \tilde{\mathbf{D}}^S$ and let $s = f_S(\mathbf{d}^S, \tilde{\mathbf{d}}^S)$ denote its deterministic state. The observable predictive signature used to align local classes determines exactly this state: for any two parent configurations,*

$$f_S(\mathbf{d}_1^S, \tilde{\mathbf{d}}_1^S) = f_S(\mathbf{d}_2^S, \tilde{\mathbf{d}}_2^S)$$
$$\iff \quad ([\mathbf{d}_1^S]_{\tilde{\mathbf{d}}_1^S}, \tilde{\mathbf{d}}_1^S)$$
$$\equiv_{\text{obs}} ([\mathbf{d}_2^S]_{\tilde{\mathbf{d}}_2^S}, \tilde{\mathbf{d}}_2^S), \qquad (15)$$

*where $[\mathbf{d}^S]_{\tilde{\mathbf{d}}^S}$ is the context-specific class from Step 2a and $\equiv_{\text{obs}}$ is the exact observational equivalence relation defined in Step 2b. This rules out both accidental merges of distinct module states and artificial splits of the same state across different shared-parent contexts.*

vii **No-Twins**. *Two distinct latent variables do not share exactly the same neighborhood (parents and children).*

viii **Maximality**. *The latent representation is not artificially over-refined: splitting any latent variable into multiple variables would violate either the Markov property of the graph or the No-Twins condition.*

**Reading Condition C.1.** Faithfulness (Condition C.1-i) is the standard requirement for recovering graphical structure from independence relations (Spirtes et al., 2000). Rank faithfulness (Condition C.1-ii) is the corresponding anti-degeneracy condition for nonnegative-rank constraints on

conditional probability tables (Huang et al., 2022). Condition C.1-iii formalizes the idea that meaningful traces occupy a tiny, structured subset of the naïve Cartesian product of token configurations. Deterministic coarsest selection (Condition C.1-iv) states that the latent selection is the coarsest statistic of its lower-level neighborhood with respect to its local Markov and predictive role in the selection hierarchy. Neighborhood coverage (Condition C.1-v) supplies a pure diagnostic cue for each latent selection and enough observed support on the other side of the rank table for the cue to reveal a genuine bottleneck. Observable module signatures (Condition C.1-vi) ensure that the predictive signatures used in the proof identify the same module state across different shared-parent contexts. Together these support and signature requirements play the role of atomic-cover conditions in latent hierarchical structure discovery (Huang et al., 2022; Dong et al., 2023). Finally, No-Twins and Maximality (Condition C.1-vii,viii) rule out redundant copies and arbitrary refinements of the latent structure, as commonly assumed in identifiability results for discrete latent variable models (Kivva et al., 2021; 2022).

**Proof roadmap.** The identification proof proceeds in three steps. First, covered local neighborhoods let us detect candidate latent states because different module choices leave different local consequences in the observed trace. Second, rank and conditional-independence comparisons identify which observed variables share the same latent parent and remove variables that are not part of the local neighborhood. Third, the one-level recovery is applied bottom-up through the hierarchy; the neighborhood coverage and no-twins conditions ensure that local pieces are stitched into a single latent structure rather than duplicated or arbitrarily split. This roadmap is the formal counterpart of the main-text assumption intuition.

**Theorem C.2** (Textual Concept Identification). *Assume the hierarchical selection process in Figure 2. Let the true parameters be $\boldsymbol{\theta}_{\mathrm{T}}$. If both the true parameterization $\boldsymbol{\theta}_{\mathrm{T}}$ and an alternative parameterization $\hat{\boldsymbol{\theta}}_{\mathrm{T}}$ satisfy Condition C.1, then equality of the induced observed distributions,*

$$P_{\boldsymbol{\theta}_{\mathrm{T}}}(\mathbf{D}) = P_{\hat{\boldsymbol{\theta}}_{\mathrm{T}}}(\mathbf{D}), \qquad (16)$$

*implies that for every level $l \in [L_{\mathrm{T}}]$, each latent concept component $\mathbf{S}_l$ is identifiable up to a component-wise relabeling of its discrete states.*

**Proof idea for Theorem C.2.** The proof below implements this roadmap constructively. At a single level, we (i) recover which observed variables are grouped together by the same latent selection using rank and conditional-independence constraints, and then (ii) recover the latent states as the *coarsest predictive partition* of the corresponding parent configurations. We then iterate this one-level identification bottom-up to recover the full hierarchy.

## C.2 Proof of Theorem C.2

**Definition C.3** (Non-negative Rank). The non-negative rank of a non-negative matrix $\mathbf{A} \in \mathbb{R}^{m \times n}$ is the smallest integer $p$ for which there exist non-negative matrices $\mathbf{B} \in \mathbb{R}^{m \times p}$ and $\mathbf{C} \in \mathbb{R}^{p \times n}$ such that $\mathbf{A} = \mathbf{BC}$.

**Lemma C.4** (Conditional Independence and Nonnegative Rank (Cohen & Rothblum, 1993)). *Let $\mathbf{P} \in \mathbb{R}^{m \times n}$ be a bivariate probability table. Then $\mathrm{rank}_+(\mathbf{P})$ equals the smallest $p$ such that $\mathbf{P}$ can be written as a convex combination of $p$ rank-one probability tables.*

**Lemma C.5** (One-level Textual Concept Identification). *Assume the hierarchical selection process Figure 2 with $L_{\mathrm{T}} = 1$. Let the true parameters be $\boldsymbol{\theta}_{\mathrm{T}}$. If both the true parameterization $\boldsymbol{\theta}_{\mathrm{T}}$ and an alternative parameterization $\hat{\boldsymbol{\theta}}_{\mathrm{T}}$ satisfy Condition C.1, then equality of the induced observed distributions $\mathbb{P}(\mathbf{D})$ implies that the latent concepts $\mathbf{S}_1$ are component-wise identifiable.*

*Proof.* We prove identifiability for $L_{\mathrm{T}} = 1$ by explicitly reconstructing (a) the bipartite adjacency between $\mathbf{S}_1$ and the observed variables $\mathbf{D}$ and (b) the discrete state of each latent component as a function of its observed neighborhood.

**Step 1: Recover the bipartite adjacency $\mathbf{D} \leftrightarrow \mathbf{S}_1$.** Fix an observed variable $D \in \mathbf{D}$. For any candidate conditioning set $\mathbf{C} \subseteq \mathbf{D} \setminus \{D\}$, define the remaining observed block $\mathbf{R} := \mathbf{D} \setminus (\{D\} \cup \mathbf{C})$ and consider the family of conditional probability tables

$$\mathbf{T}_{D,\mathbf{R}}^{(\mathbf{C}=\mathbf{c})} := \mathbb{P}(D, \mathbf{R} \mid \mathbf{C} = \mathbf{c}), \qquad \mathbf{c} \in \mathrm{supp}(\mathbf{C}). \quad (17)$$

*Covered anchor variables create an informative rank drop.* Suppose $D$ is a pure parent for some latent variable $S \in \mathbf{S}_1$ as in Condition C.1-v. If we choose $\mathbf{C}$ to include all other observed parents of $S$ (i.e., $\mathbf{C} \supseteq \mathrm{CoPa}(D)$), then, conditioned on $\mathbf{C}$, the dependence between $D$ and $\mathbf{R}$ factors through the finite latent variable $S$. By Lemma C.4, each table $\mathbf{T}_{D,\mathbf{R}}^{(\mathbf{C}=\mathbf{c})}$ is then a mixture of $|\mathrm{supp}(S)|$ rank-one tables, so

$$\mathrm{rank}_+\left(\mathbf{T}_{D,\mathbf{R}}^{(\mathbf{C}=\mathbf{c})}\right) \leq |\mathrm{supp}(S)| < |\mathrm{supp}(D)|, \quad (18)$$

where the strict inequality uses Condition C.1-v. When $\mathbf{C} = \mathrm{CoPa}(D)$, the rank-table support part of the same condition ensures that the comparison block $\mathbf{R}$ has enough support for this strict inequality to be an informative bottleneck rather than a dimensional ceiling of the table. Thus, anchors are detectable from the observed law via a strict nonnegative-rank drop.

*Minimal separating sets identify co-parents.* Define $\widehat{\mathrm{CoPa}(D)}$ as any *inclusion-minimal* set $\mathbf{C} \subseteq \mathbf{D} \setminus \{D\}$ for which there exists $\mathbf{c} \in \mathrm{supp}(\mathbf{C})$ such that

$$\mathrm{rank}_+\left(\mathbf{T}_{D,\mathbf{R}}^{(\mathbf{C}=\mathbf{c})}\right) < |\mathrm{supp}(D)|. \quad (19)$$

We claim that for a pure parent $D$ this minimal set coincides with the true co-parent set: $\widehat{\mathrm{CoPa}}(D) = \mathrm{CoPa}(D)$.

First, $\mathrm{CoPa}(D) \subseteq \widehat{\mathrm{CoPa}}(D)$. If some co-parent were omitted from $\mathbf{C}$, then (by faithfulness) there would remain an active path between $D$ and that omitted variable given $\mathbf{C}$. If the strict rank drop in (19) still held after omitting such a co-parent, that nonnegative-rank constraint would be an accidental constraint not entailed by the latent DAG with the stated cardinalities, contradicting rank faithfulness in Condition C.1-ii.

Second, $\widehat{\mathrm{CoPa}}(D) \subseteq \mathrm{CoPa}(D)$. Any variable outside $\mathrm{CoPa}(()D)$ is d-separated from $D$ once we condition on the true co-parent set and the associated latent child, hence removing such an extraneous variable from $\mathbf{C}$ cannot destroy the rank drop. By the inclusion-minimality in the definition of $\widehat{\mathrm{CoPa}}(D)$, no such extraneous variable can appear.

Repeating this procedure over observed variables detects a covered anchor for each latent node and identifies which observed variables share that latent neighbor. Condition C.1-vii (No-Twins) ensures that the resulting grouping defines a unique latent node for each distinct neighborhood, yielding the bipartite adjacency between $\mathbf{D}$ and $\mathbf{S}_1$.

**Step 2: Recover the latent state as a predictive partition.** Fix a latent variable $S \in \mathbf{S}_1$ and write its observed parents as

$$\mathrm{Pa}(S) = \mathbf{D}^S \cup \tilde{\mathbf{D}}^S, \tag{20}$$

where $\mathbf{D}^S$ are the pure parents (anchors) and $\tilde{\mathbf{D}}^S$ are the remaining (shared) parents. Let $\mathbf{U} := \mathbf{D} \setminus \mathrm{Pa}(S)$ denote the observed variables outside the neighborhood of $S$. The Markov property for the latent graph implies the conditional independence

$$\mathbf{D}^S \perp\!\!\!\perp \mathbf{U} \mid (S, \tilde{\mathbf{D}}^S). \tag{21}$$

*Step 2a: merge anchor configurations inside a fixed shared-parent context.* Fix any $\tilde{\mathbf{d}}^S \in \mathrm{supp}(\tilde{\mathbf{D}}^S)$. Define an equivalence relation on anchor assignments by

$$\mathbf{d}_1^S \sim_{\tilde{\mathbf{d}}^s} \mathbf{d}_2^S \iff$$
$$\mathbb{P}\left(\mathbf{U} \mid \mathbf{D}^S = \mathbf{d}_1^S, \tilde{\mathbf{D}}^S = \tilde{\mathbf{d}}^s\right) = \tag{22}$$
$$\mathbb{P}\left(\mathbf{U} \mid \mathbf{D}^S = \mathbf{d}_2^S, \tilde{\mathbf{D}}^S = \tilde{\mathbf{d}}^s\right).$$

By (21), the conditional law of $\mathbf{U}$ depends on $\mathbf{D}^S$ only through the latent value $S$ when $\tilde{\mathbf{D}}^S$ is fixed. Hence each equivalence class corresponds to a context-specific latent-state label. For an anchor assignment $\mathbf{d}^S$ in the fixed context $\tilde{\mathbf{d}}^S$, let $[\mathbf{d}^S]_{\tilde{\mathbf{d}}^s}$ denote the resulting class index.

*Step 2b: align these class indices across different shared-parent contexts.* The shared parents $\tilde{\mathbf{D}}^S$ may also be parents

of other latent variables. Let $\mathrm{Ch}(\tilde{\mathbf{D}}^S)$ denote all latent children of the shared-parent set, and define the additional observed parents of these children by

$$\tilde{\tilde{\mathbf{D}}}^S := \mathrm{Pa}(\mathrm{Ch}(\tilde{\mathbf{D}}^S)) \setminus \{\mathbf{D}^S, \tilde{\mathbf{D}}^S\}. \tag{23}$$

Conditioning on $\mathrm{Ch}(\tilde{\mathbf{D}}^S)$ and their other observed parents blocks all paths from $(\mathbf{D}^S, \tilde{\mathbf{D}}^S)$ to variables outside their joint neighborhood, yielding

$$(\mathbf{D}^S, \tilde{\mathbf{D}}^S) \perp\!\!\!\perp \mathbf{D} \setminus \mathrm{Pa}(\mathrm{Ch}(\tilde{\mathbf{D}}^S)) \mid (\mathrm{Ch}(\tilde{\mathbf{D}}^S), \tilde{\tilde{\mathbf{D}}}^S). \tag{24}$$

We now align the context-specific labels across different shared-parent values by comparing the observable conditional laws implied by (24). Concretely, we define exact observational equivalence by declaring

$$([\mathbf{d}_1^S]_{\tilde{\mathbf{d}}_1^S}, \tilde{\mathbf{d}}_1^S) \equiv_{\mathrm{obs}} ([\mathbf{d}_2^S]_{\tilde{\mathbf{d}}_2^S}, \tilde{\mathbf{d}}_2^S) \tag{25}$$

if for every $\tilde{\tilde{\mathbf{d}}}^S \in \mathrm{supp}(\tilde{\tilde{\mathbf{D}}}^S)$,

$$\mathbb{P}\left(\mathbf{D} \setminus \mathrm{Pa}(\mathrm{Ch}(\tilde{\mathbf{D}}^S)) \mid ([\mathbf{D}^S]_{\tilde{\mathbf{D}}^s}, \tilde{\mathbf{D}}^S) = ([\mathbf{d}_1^S]_{\tilde{\mathbf{d}}_1^S}, \tilde{\mathbf{d}}_1^S), \tilde{\tilde{\mathbf{D}}}^S = \tilde{\tilde{\mathbf{d}}}^s\right)$$
$$= \mathbb{P}\left(\mathbf{D} \setminus \mathrm{Pa}(\mathrm{Ch}(\tilde{\mathbf{D}}^S)) \mid ([\mathbf{D}^S]_{\tilde{\mathbf{D}}^s}, \tilde{\mathbf{D}}^S) = ([\mathbf{d}_2^S]_{\tilde{\mathbf{d}}_2^S}, \tilde{\mathbf{d}}_2^S), \tilde{\tilde{\mathbf{D}}}^S = \tilde{\tilde{\mathbf{d}}}^s\right). \tag{26}$$

By Condition C.1-vi, this observable equivalence relation coincides with equality of the true deterministic state of $S$. Let $[(\mathbf{D}^S, \tilde{\mathbf{D}}^S)]_{\mathrm{obs}}$ denote the resulting equivalence class. We define the recovered latent variable $\hat{S}$ as the class index

$$\hat{S} := \hat{f}_S(\mathbf{D}^S, \tilde{\mathbf{D}}^S) := [(\mathbf{D}^S, \tilde{\mathbf{D}}^S)]_{\mathrm{obs}}. \tag{27}$$

By construction, $\hat{S}$ is a deterministic function of the observed neighborhood.

**Step 3: $\hat{S}$ matches $S$ up to relabeling.** Condition C.1-vi states that the observational equivalence classes constructed in Step 2b are exactly the level sets of the true deterministic map $f_S(\mathbf{D}^S, \tilde{\mathbf{D}}^S)$. Thus two parent configurations receive the same recovered state if and only if they induce the same true state of $S$. The recovered partition therefore coincides with the true partition of parent configurations, which implies that $\hat{S}$ equals $S$ up to a bijection on its discrete state labels.

Applying the above construction to each component of $\mathbf{S}_1$ completes the one-level identifiability claim. $\qquad\square$

**Condition C.1** (Textual Concept Identification Conditions).

  *i* **Faithfulness.** *All (conditional) independence relations in the observed distribution are entailed by d-separation in the underlying graph.*

  *ii* **Rank faithfulness.** *The observed distribution $P$ is rank-faithful to the latent DAG $G$: every nonnegative-rank constraint on a conditional sub-probability table*

*of observed variables that holds in $P$ is entailed by the class of discrete structural models Markov to $G$ with the stated latent cardinalities.*

iii **Restricted choice sets**. *Each selection variable $S_l$ only realizes a strict subset of the combinatorial state space obtained by freely composing its immediate constituents. Concretely,*

$$\operatorname{supp} S_l \subsetneq f_{\mathbf{D} \to S_l}\big(\Omega^{n(\operatorname{Pa}(S_l))}\big), \qquad (11)$$

*where $f_{\mathbf{D} \to S_l}$ denotes the (level-$l$) composition map from lower-level variables to $S_l$.*

iv **Deterministic coarsest selection**. *For each latent node $S$, $S$ is the unique coarsest deterministic statistic of $\operatorname{Pa}(S)$.*

v **Neighborhood coverage**. *For every selection variable $S_l$, there exists a diagnostic pure parent $D^\star \in \operatorname{Pa}(S_l)$ such that $\operatorname{Ch}(D^\star) = \{S_l\}$ and*

$$|\operatorname{supp} D^\star| > |\operatorname{supp} S_l|. \qquad (12)$$

*Moreover, the observed variables outside this diagnostic cue provide a non-vacuous comparison table. Writing*

$$C_{D^\star} := \operatorname{CoPa}(D^\star), \qquad R_{D^\star} := \mathbf{D} \setminus \big(\{D^\star\} \cup C_{D^\star}\big), \qquad (13)$$

*there exists $c \in \operatorname{supp}(C_{D^\star})$ such that*

$$\big|\operatorname{supp}\big(R_{D^\star} \mid C_{D^\star} = c\big)\big| \geq |\operatorname{supp}(S_l)|. \qquad (14)$$

*Thus each latent selection has a pure local cue, and the opposite side of the rank table is large enough for a rank drop to reveal a genuine latent bottleneck.*

vi **Observable module signatures**. *For each selection variable $S$, write $\operatorname{Pa}(S) = \mathbf{D}^S \cup \tilde{\mathbf{D}}^S$ and let $s = f_S(\mathbf{d}^S, \tilde{\mathbf{d}}^S)$ denote its deterministic state. The observable predictive signature used to align local classes determines exactly this state: for any two parent configurations,*

$$f_S(\mathbf{d}_1^S, \tilde{\mathbf{d}}_1^S) = f_S(\mathbf{d}_2^S, \tilde{\mathbf{d}}_2^S)$$
$$\iff \quad ([\mathbf{d}_1^S]_{\tilde{\mathbf{d}}_1^S}, \tilde{\mathbf{d}}_1^S)$$
$$\equiv_{\operatorname{obs}} ([\mathbf{d}_2^S]_{\tilde{\mathbf{d}}_2^S}, \tilde{\mathbf{d}}_2^S), \qquad (15)$$

*where $[\mathbf{d}^S]_{\tilde{\mathbf{d}}^S}$ is the context-specific class from Step 2a and $\equiv_{\operatorname{obs}}$ is the exact observational equivalence relation defined in Step 2b. This rules out both accidental merges of distinct module states and artificial splits of the same state across different shared-parent contexts.*

vii **No-Twins**. *Two distinct latent variables do not share exactly the same neighborhood (parents and children).*

viii **Maximality**. *The latent representation is not artificially over-refined: splitting any latent variable into multiple variables would violate either the Markov property of the graph or the No-Twins condition.*

**Theorem C.2** (Textual Concept Identification). *Assume the hierarchical selection process in Figure 2. Let the true parameters be $\boldsymbol{\theta}_{\mathrm{T}}$. If both the true parameterization $\boldsymbol{\theta}_{\mathrm{T}}$ and an alternative parameterization $\hat{\boldsymbol{\theta}}_{\mathrm{T}}$ satisfy Condition C.1, then equality of the induced observed distributions,*

$$P_{\boldsymbol{\theta}_{\mathrm{T}}}(\mathbf{D}) = P_{\hat{\boldsymbol{\theta}}_{\mathrm{T}}}(\mathbf{D}), \qquad (16)$$

*implies that for every level $l \in [L_{\mathrm{T}}]$, each latent concept component $\mathbf{S}_l$ is identifiable up to a component-wise relabeling of its discrete states.*

*Proof.* We lift Lemma C.5 to the full hierarchy via induction over levels. At the bottom, take the observed trace variables $\mathbf{D}$ as the level-$(L_{\mathrm{T}} + 1)$ "inputs". Lemma C.5 identifies the first latent layer $\mathbf{S}_{L_{\mathrm{T}}}$ (up to component-wise relabeling) together with its adjacency to $\mathbf{D}$. Because Condition C.1-iv identifies each recovered latent layer as a sample-level deterministic statistic of the layer below, the recovered $\mathbf{S}_{L_{\mathrm{T}}}$ can be treated as observed after relabeling. Once $\mathbf{S}_{L_{\mathrm{T}}}$ is identified, we can treat it as observed and apply the same argument to identify $\mathbf{S}_{L_{\mathrm{T}}-1}$, and so on. Iterating up to level 1 yields identifiability of every layer $\mathbf{S}_l$. $\square$

### C.3 Proof of Theorem 3.4

*Proof.* We prove that the local witness condition in Theorem 3.4 is sufficient for composing locally witnessed constraints under a compositionally novel prompt.

**Setup and notation.** Fix $p_{\operatorname{new}} \in \Omega_{\operatorname{comp}}$. Recall the training-witnessed local compatibility set from (6): for a node $U$ and a child tuple $v_{\operatorname{Ch}(U)}$, $\mathcal{W}_U(v_{\operatorname{Ch}(U)})$ is the set of parent values $u$ such that the assignment $(U = u, \operatorname{Ch}(U) = v_{\operatorname{Ch}(U)})$ occurs with nonzero probability under some training prompt $p \in \Omega_{\operatorname{supp}}$. Fix a high-level latent configuration in the support of $p_{\operatorname{new}}$ and consider the full descendant assignment induced by the hierarchical selection model. All local tuples below are therefore induced by this single global assignment, rather than chosen independently across nodes.

**Composing witnesses layer by layer.** Because the selection hierarchy is acyclic and layered (Figure 2), we can construct lower levels one layer at a time.

Consider any node $U$ whose children have already been assigned a tuple $v_{\operatorname{Ch}(U)}$. Let $u$ be the value of $U$ induced by the same fixed assignment. If this local family can arise under $p_{\operatorname{new}}$, then by the local witness condition (10) we have

$$u \in \mathcal{W}_U\big(v_{\operatorname{Ch}(U)}\big). \qquad (28)$$

By the definition of $\mathcal{W}_U$, this value $u$ has been witnessed together with the *full* child tuple $v_{\mathrm{Ch}(U)}$ under some training prompt, so it is simultaneously compatible with every child of $U$.

Proceeding recursively over all nodes in descending layer order yields a full assignment in which every local parent–children family is witnessed on the training support. Thus the novel prompt does not require any genuinely new local interface beyond what training has already exercised.

**Implication for learned models.** Any learned model that recovers the local witness relations $\mathcal{W}_U(\cdot)$ from training can realize the above construction by selecting, at each node $U$, the induced value $u \in \mathcal{W}_U(v_{\mathrm{Ch}(U)})$. Hence it can form a consistent composition under $p_{\mathrm{new}}$. □

## D   Formal Support View of SFT and RL

This section formalizes the support-level intuition behind Proposition 3.3. We prove two complementary facts. First, SFT can leave an unfalsifiable support blind spot: if demonstrations reveal only a subset of valid traces, then an SFT-only learner cannot distinguish the true trace law from an alternative law that treats the visible subset as complete. Second, RL can make useful hidden events statistically visible when rollouts can reach them and the final-answer reward distinguishes them from unhelpful variants.

**Setup.** Fix a prompt set $\mathcal{C}_0$ with prompt marginal $\mu^\star$ and true prompt-trace law

$$Q^\star(p, d) = \mu^\star(p)\, p^\star(d \mid p). \tag{29}$$

For each prompt $p$, suppose SFT reveals only a subset $A_S(p)$ of valid traces, with true mass

$$s(p) := \sum_d p^\star(d \mid p)\mathbf{1}\{d \in A_S(p)\}. \tag{30}$$

The SFT-visible conditional is

$$q_S(d \mid p) := \frac{p^\star(d \mid p)\mathbf{1}\{d \in A_S(p)\}}{s(p)}. \tag{31}$$

We assume $s(p) > 0$ on the prompts under consideration. RL rollouts receive a binary final-answer reward $R \in \{0, 1\}$.

**Identifiability-critical events.** Our identification and composition conditions require the learner to observe specific trace-level configurations. Examples include a module in a particular neighboring context, as in Condition C.1-v, or a parent value co-occurring with a specific child tuple, as in the local witness condition of Theorem 3.4. As in the main text, we write such a local event as $\mathcal{E} = \{U = u, \mathrm{Ch}(U) = v_{\mathrm{Ch}(U)}\}$.

**Theorem D.1** (SFT non-identifiability under hidden trace support)**.** *Define the alternative law*

$$\widetilde{Q}(p, d) := \mu^\star(p)\, q_S(d \mid p), \tag{32}$$

*which treats the SFT-visible traces as the full conditional trace distribution. Then:*

*(i) $Q^\star$ and $\widetilde{Q}$ induce identical SFT observations.*

*(ii) Their trace laws differ by the hidden trace mass,*

$$\|Q^\star - \widetilde{Q}\|_1 = 2\,\mathbb{E}_{\mu^\star}[1 - s(p)]. \tag{33}$$

*(iii) Any identifiability-critical event that occurs only outside $A_S(p)$ is absent under $\widetilde{Q}$ and therefore cannot be certified from SFT observations alone.*

*Proof.* Under $Q^\star$, the SFT observation process reveals traces according to $q_S(d \mid p)$. Under $\widetilde{Q}$, the full conditional trace law is exactly $q_S(d \mid p)$. Therefore both laws produce the same supervised observations for every prompt, proving (i).

For (ii), condition on a fixed prompt $p$ and compute the $\ell_1$ distance between the true conditional and the visible conditional:

$$\sum_d |p^\star(d \mid p) - q_S(d \mid p)|$$
$$= \sum_{d \in A_S(p)} p^\star(d \mid p)\left|\frac{1}{s(p)} - 1\right| + \sum_{d \notin A_S(p)} p^\star(d \mid p)$$
$$= (1 - s(p)) + (1 - s(p)) = 2(1 - s(p)). \tag{34}$$

Averaging over $p \sim \mu^\star$ gives the claimed identity.

For (iii), if an event $\mathcal{E}$ occurs only on traces outside $A_S(p)$, then $q_S(\mathcal{E} \mid p) = 0$ and therefore $\widetilde{Q}(\mathcal{E} \mid p) = 0$. Since SFT observations are identical under $Q^\star$ and $\widetilde{Q}$, an SFT-only test cannot tell whether $\mathcal{E}$ is truly present in hidden valid traces or absent from the trace law. If such events are required for the neighborhood coverage or local witness conditions, the obstruction is non-identifiability rather than finite-sample error. □

**Interpretation.** Even with infinite SFT data, the learner cannot distinguish $Q^\star$ from $\widetilde{Q}$ using supervised observations alone. The mass $1 - s(p)$ consists of within-prompt traces that SFT never reveals. If identifiability-critical configurations live in that hidden mass, no SFT-only learner can certify the module neighborhoods or interfaces needed by the theory.

**Theorem D.2** (RL enrichment via verifiable reward)**.** *Let $\hat{p}_0(\mathbf{D} \mid \mathbf{P})$ denote the current rollout model before the local update. In this appendix, $\hat{p}_0(\mathcal{E} \mid p)$ and $\mathrm{Pr}(R = 1 \mid p)$*

*abbreviate conditioning on* $\mathbf{P} = p$. *For an identifiability-critical event* $\mathcal{E}$, *define its within-prompt reward gap*

$$\Delta_{\mathcal{E}}(p) := \Pr(R = 1 \mid \mathcal{E}, p) - \Pr(R = 1 \mid \mathcal{E}^c, p). \quad (35)$$

*If* $0 < \hat{p}_0(\mathcal{E} \mid p) < 1$, $\Delta_{\mathcal{E}}(p) > 0$, *and* $\Pr(R = 1 \mid p) > 0$, *then reward-positive rollouts enrich* $\mathcal{E}$:

$$\Pr(\mathcal{E} \mid R = 1, p) - \hat{p}_0(\mathcal{E} \mid p)$$
$$= \frac{\hat{p}_0(\mathcal{E} \mid p)\big(1 - \hat{p}_0(\mathcal{E} \mid p)\big)\Delta_{\mathcal{E}}(p)}{\Pr(R = 1 \mid p)} > 0. \quad (36)$$

*Moreover, a local policy-gradient update that increases the log-probability of traces containing* $\mathcal{E}$ *has positive expected reward gradient*

$$\frac{d}{d\theta} J(\theta)\bigg|_{\theta=0} = \mathbb{E}_{p \sim \mu^\star} \big[\hat{p}_0(\mathcal{E} \mid p)\big(1 - \hat{p}_0(\mathcal{E} \mid p)\big)\Delta_{\mathcal{E}}(p)\big], \quad (37)$$

*where* $J(\theta)$ *is the expected final-answer reward under the locally tilted rollout law*

$$\hat{p}_\theta(d \mid p) := \frac{\hat{p}_0(d \mid p) \exp(\theta \mathbf{1}\{d \in \mathcal{E}\})}{\sum_{d'} \hat{p}_0(d' \mid p) \exp(\theta \mathbf{1}\{d' \in \mathcal{E}\})}. \quad (38)$$

*Proof.* Let

$$a(p) := \Pr(R = 1 \mid \mathcal{E}, p), \quad (39)$$
$$b(p) := \Pr(R = 1 \mid \mathcal{E}^c, p). \quad (40)$$

Then $\Delta_{\mathcal{E}}(p) = a(p) - b(p)$, and the prompt-level reward probability is

$$\Pr(R = 1 \mid p) = \hat{p}_0(\mathcal{E} \mid p)a(p) + \big(1 - \hat{p}_0(\mathcal{E} \mid p)\big)b(p). \quad (41)$$

Bayes' rule gives

$$\Pr(\mathcal{E} \mid R = 1, p) = \frac{\hat{p}_0(\mathcal{E} \mid p)a(p)}{\Pr(R = 1 \mid p)}. \quad (42)$$

Subtracting $\Pr(\mathcal{E} \mid p) = \hat{p}_0(\mathcal{E} \mid p)$ yields

$$\Pr(\mathcal{E} \mid R = 1, p) - \hat{p}_0(\mathcal{E} \mid p)$$
$$= \frac{\hat{p}_0(\mathcal{E} \mid p)}{\Pr(R = 1 \mid p)}\Big(a(p) - \hat{p}_0(\mathcal{E} \mid p)a(p)$$
$$- \big(1 - \hat{p}_0(\mathcal{E} \mid p)\big)b(p)\Big)$$
$$= \frac{\hat{p}_0(\mathcal{E} \mid p)\big(1 - \hat{p}_0(\mathcal{E} \mid p)\big)}{\Pr(R = 1 \mid p)}\big(a(p) - b(p)\big)$$
$$= \frac{\hat{p}_0(\mathcal{E} \mid p)\big(1 - \hat{p}_0(\mathcal{E} \mid p)\big)}{\Pr(R = 1 \mid p)}\Delta_{\mathcal{E}}(p). \quad (43)$$

This proves enrichment and also shows that positive and negative traces provide a contrastive signal whenever $\Delta_{\mathcal{E}}(p) > 0$.

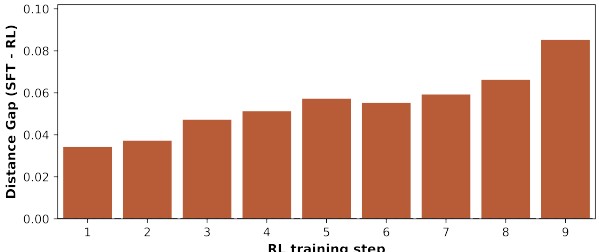

*Figure 9.* **Toy binary-output diagnostic for trace-support recovery.** In this controlled diagnostic, RL rollout feedback reduces the distance to the known target distribution relative to the SFT-only model.

For the policy-gradient statement, differentiate the local tilt at $\theta = 0$:

$$\frac{d}{d\theta} \log \hat{p}_\theta(d \mid p)\bigg|_{\theta=0} = \mathbf{1}\{d \in \mathcal{E}\} - \hat{p}_0(\mathcal{E} \mid p). \quad (44)$$

For a fixed prompt, write $J_p(\theta) := \mathbb{E}_{d \sim \hat{p}_\theta(\cdot \mid p), R}[R]$. The REINFORCE identity gives

$$\frac{d}{d\theta} J_p(\theta)\bigg|_{\theta=0} = \mathbb{E}_{d \sim \hat{p}_0(\cdot \mid p), R} \big[R\big(\mathbf{1}\{d \in \mathcal{E}\} - \hat{p}_0(\mathcal{E} \mid p)\big) \mid p\big]$$
$$= \mathrm{Cov}\big(R, \mathbf{1}\{d \in \mathcal{E}\} \mid p\big)$$
$$= \hat{p}_0(\mathcal{E} \mid p)\big(1 - \hat{p}_0(\mathcal{E} \mid p)\big)$$
$$\times \Delta_{\mathcal{E}}(p). \quad (45)$$

Averaging over $p \sim \mu^\star$ gives the displayed gradient formula. Thus, when $\mathcal{E}$ is reachable and reward-informative, RL has a positive local ascent direction for increasing the probability of traces containing $\mathcal{E}$. $\qquad\square$

**When RL has an advantage over SFT.** The two results identify three governing factors. First, *censorship severity*: larger hidden mass $1 - s(p)$ creates a wider SFT non-identifiability gap. Second, *reachability*: if $\hat{p}_0(\mathcal{E} \mid p) = 0$, rollouts never visit the missing event and RL cannot make it statistically visible. Third, *reward informativeness*: if $\Delta_{\mathcal{E}}(p) = 0$, reward does not distinguish traces containing $\mathcal{E}$ from traces omitting it, so RL reduces to undirected exploration for that event. When all three align, RL can expose and upweight useful local configurations that SFT demonstrations leave hidden.

# E   Toy Diagnostic for Trace-Support Recovery

The SFT training data is described below. The input sequence $X$ has a length of $S$, and each input character $x_i$ has a value range of $\{0, 1\}$. There are $N$ input sequences, $N < 2^S$. Each input character $x_i$ corresponds to an output sequence $Y_i$ of length $D$, with output characters having a

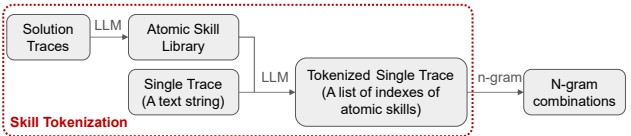

*Figure 10.* **Skill tokenization pipeline.** This pipeline maps solution traces to sequences of atomic skill tokens and derives n-gram combinations.

value range of $\{0, 1\}$. Given the value of $x_i$, there are $M$ candidate values for $Y_i$, $M < 2^D$. The complete input sequence determines the value of the last character $Y_{\text{end}}$ in the output sequence, with $Y_{\text{end}}$ having a value range of $\{0, 1\}$. The complete output sequence $Y$ is formed by concatenating all $Y_i$ and then appending $Y_{\text{end}}$. Therefore, the length of the complete output sequence $Y$ is $SD + 1$.

Based on the SFT-trained model, RL training is performed. RL training uses the same input as the SFT training phase. All $Y_i$ are explored by the model itself. The reward calculation logic is based on the $(SD + 1)$-th character output by the model. The details of the reward calculation are as follows. Obtain the probability distribution of the $(SD + 1)$-th character in the vocabulary space from the model's output. Observe the probability values of characters 0 and 1, select the one with the higher probability, and compare it with the correct answer. If they are the same, the reward is 1; otherwise, the reward is 0.

After RL training, proceed to the evaluation phase. The input used for evaluation is the same as that used in the training phase. Given an input, the distribution of the correct answer is $\text{Pr}_{\text{true}}$, the output distribution of the model trained by SFT is $\text{Pr}_{\text{SFT}}$, and the output distribution of the model trained by RL is $\text{Pr}_{\text{RL}}$. Note that we focus only on characters 0 and 1 in the vocabulary space. Calculate the distance between $\text{Pr}_{\text{true}}$ and $\text{Pr}_{\text{SFT}}$, and calculate the distance between $\text{Pr}_{\text{true}}$ and $\text{Pr}_{\text{RL}}$.

Specifically, the setting is: $S = 4, N = 192, D = 2, M = 2$. Figure 9 shows that, in this toy diagnostic, the RL-trained model's output distribution is closer to the known target distribution than the SFT-trained model's output distribution.

## F   Full Experimental Setup

This section gives the full experimental setup in Table 3.

## G   Details of Experiments Related to Real-Model Evidence

This section provides the full pipeline for the real-model evidence summarized in Section 4.6.

First, we collect solution traces from real SFT and RL models. Specifically, we sample examples from established

mathematics benchmarks, including MATH-500 (Lightman et al., 2023), AIME 2024 (American Institute of Mathematics, 2024), AIME 2025 (American Institute of Mathematics, 2025), AMC (Mathematical Association of America, 2023), and GSM8K (Cobbe et al., 2021), yielding 643 data points in total. We then use Qwen3-4B (Yang et al., 2025) and Qwen3-4B-Thinking-2507 to generate solution traces for each example.

Next, as illustrated in Figure 10, we construct a skill tokenization method for solution traces produced by real post-trained LLMs. Specifically, we use a strong LLM to summarize a library of atomic mathematical skills from the full set of traces. We assign each atomic skill in this library a unique index (a positive integer). With the assistance of the same LLM, we tokenize each solution trace (a natural-language text string) into a sequence of indices corresponding to the atomic skills it contains.

Based on the tokenized traces, we apply an $n$-gram algorithm to extract $n$-grams ($n = 2, 3, \ldots$). The resulting frequency statistics provide, for each $n$-gram, its occurrence frequency under both the SFT model (Qwen3-4B) and the RL model (Qwen3-4B-Thinking-2507).

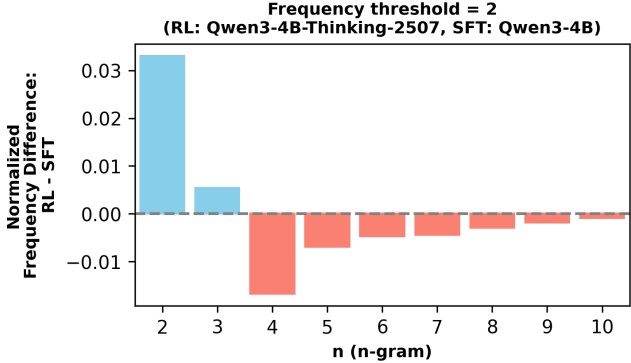

*Figure 11.* **Normalized n-gram frequency difference between the RL and SFT model.** Positive values indicate higher relative frequency in the RL model (RL - SFT), whereas negative values indicate higher relative frequency in the SFT model.

We visualize the differences in $n$-gram frequencies between the two models in Figure 11. The results indicate that:

- The SFT-trained model has relatively higher frequency on longer skill sequences, consistent with reuse of more fixed trace templates.

- The RL-trained model has relatively higher frequency on shorter skill combinations, consistent with local recombination of reusable modules.

Figure 12 compares the number of unique $n$-grams produced by the two models.

*Table 3.* **Controlled-experiment setup.** Main-text experiments use the same prompt family for SFT and RL while changing which atomic modules and compositions appear in each stage.

| Item | Setting |
|------|---------|
| Base model | Llama-3.1-8B-Instruct (Dubey et al., 2024) |
| Task family | Synthetic string transformations with 24 atomic skills from (Yuan et al., 2025) and 10 routing mechanisms |
| SFT data | Correct reasoning traces; ten sampled reasoning-inclusive responses per problem, retaining correct final answers |
| SFT training | 2 epochs, learning rate $2 \times 10^{-5}$, batch size 128 |
| RL reward | Final string exact-match reward; no step-level labels are used |
| RL rollouts | Multiple rollouts per prompt, retaining both successful and unsuccessful samples for reward-based updates |
| Default RL settings | Learning rate $1 \times 10^{-6}$, maximum length 8192, temperature 1.0, rollout batch size 16 |
| Regularization and filtering | KL coefficient 0, entropy coefficient 0, filtering prompts whose rollouts are all correct or all incorrect |
| Evaluation | Accuracy on seen compositions, unseen compositions, and depth extrapolation across composition levels |

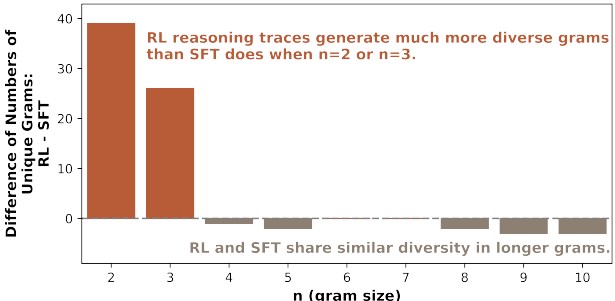

*Figure 12.* **Difference of numbers of unique grams between the RL and SFT model.** We calculate the number of unique grams for both RL and SFT models, then visualize their differences here.

- RL reasoning traces generate substantially more diverse $n$-grams when $n = 2$ and $n = 3$.

- RL and SFT reasoning traces have similar diversity for longer $n$-grams.

The short-$n$-gram difference is the key diagnostic: it shows that RL expands the set of local skill combinations available for later recombination. Similar diversity for longer $n$-grams does not contradict the depth-extrapolation gains in the controlled experiments, because the proposed mechanism depends on recovering local modules and interfaces rather than enumerating every long trace template. Overall, these observations support a reusable library of atomic modules rather than reliance on a single long template.

## H Recovery Effect of Re-injection Depends on the Training Stage

In Figure 5, all settings share the same SFT model, which is trained on data with depth $L = 3$. Under the base RL configuration ($L = 3$), we first remove from the RL training corpus all composed instances that involve $\mathbf{f}_\star$. We then re-inject $\mathbf{f}_\star$ (Isolated, $L = 1$) by augmenting the remaining RL data with isolated examples of $\mathbf{f}_\star$. This intervention leads to a measurable, albeit partial, recovery in performance.

In addition to Figure 5, we perform a complementary experiment to assess whether the recovery effect depends on the

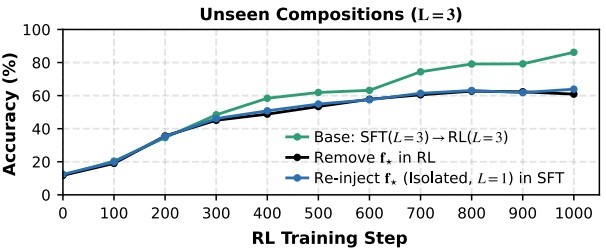

*Figure 13.* **Recovery effect of re-injecting $\mathbf{f}_\star$ ($L = 1$) in the SFT stage.** We compare the SFT→RL baseline with variants that remove or re-inject $\mathbf{f}_\star$.

training stage at which $\mathbf{f}_\star$ is reintroduced. Concretely, we apply the same re-injection strategy—adding $\mathbf{f}_\star$ (Isolated, $L = 1$)—to the SFT dataset rather than to the RL dataset. The corresponding results are reported in Figure 13. In contrast to RL-stage re-injection, incorporating $\mathbf{f}_\star$ into SFT yields negligible improvement, indicating that it does not effectively restore the lost performance. Overall, these results suggest that the efficacy of re-injecting new information is highly stage-dependent, with substantially stronger effects when the re-injection is performed during RL than during SFT.

## I Training Details

This section gives a description of the synthetic data-generating process (DGP), and the SFT/RL training protocol. The goal is to make the source of post-training data explicit: the latent variables in the DGP are exactly the atomic skill choices and routing choices that our theory treats as reusable modules.

**Atomic module library.** We use a controlled string-transformation environment. Let $\mathcal{F} = \{f_1, \ldots, f_{24}\}$ denotes the set of deterministic string transformation functions reused from (Yuan et al., 2025). Each $f_i$ maps a string, or a fixed finite tuple of strings/auxiliary arguments, to a string. Whenever a transformation has auxiliary arguments, we treat the instantiated transformation as the atomic skill used in the composition. These functions serve as *atomic skills*.

We additionally define a routing library $\mathcal{R}$ of 10 deterministic input-construction mechanisms. A router specifies which previous intermediate value(s) are supplied to the next skill and how they are combined. For example, at the third step,

$$\rho_1^{(3)}(y_1, y_2) = y_1, \qquad \rho_2^{(3)}(y_1, y_2) = \mathrm{concat}(y_1, y_2),$$

represent two different routing mechanisms: the next skill receives either the first intermediate value alone or the concatenation of the first two intermediate values.

**Data-generating process.** A compositional instance is generated from a *composition signature*

$$\sigma = \big((i_1, \rho_1), \ldots, (i_L, \rho_L)\big),$$

where $L$ is the composition depth, $i_t \in \{1, \ldots, 24\}$ indexes the skill used at step $t$, and $\rho_t \in \mathcal{R}$ indexes the router used to form the input to that skill. The level $L$ is therefore the number of atomic skill applications in the trace. For each split $s \in \{\mathrm{SFT}, \mathrm{RL}, \mathrm{eval}\}$, we define a split-specific support $\mathcal{C}_s$ over composition signatures. This support is the object we intervene on in the experiments: for example, we remove all signatures containing a held-out skill or router, re-inject selected signatures, or impose different relationships between $\mathcal{C}_{\mathrm{SFT}}$ and $\mathcal{C}_{\mathrm{RL}}$.

Given a split $s$, a data point is generated as follows. First, sample a composition signature $\sigma \sim \mathrm{Unif}(\mathcal{C}_s)$ and an input string $x$ of length between 3 and 10. Then compute the intermediate strings recursively:

$$y_0 = x, \;\; u_t = \rho_t(x, y_0, y_1, \ldots, y_{t-1}),$$
$$y_t = f_{i_t}(u_t), \quad t = 1, \ldots, L.$$

The gold final answer is the deterministic string

$$Y(\sigma, x) = y_L.$$

The model prompt contains only the code-like composition expression, the function identifiers, and the input string; it does *not* contain the function definitions during SFT, RL, or evaluation. Thus, for a signature $\sigma$, the prompt has the form

$$P_{\sigma,x} = \mathtt{def\ main\_solution(x):\ \ return\ } e_\sigma(x),$$

where $e_\sigma$ is the nested/routed expression induced by $\sigma$. The target response is required to place the final string in a JSON field, e.g.,

$$\{\mathtt{"output":}\ \ Y(\sigma, x)\}.$$

This DGP makes the latent structure known to the experimenter but hidden from the learner. The latent skill variables are the indices $i_t$; the latent routing variables are the indices $\rho_t$; the observed trace $D$ is the model's textual reasoning trajectory and final answer. Therefore, OOD evaluation is performed by holding out *composition signatures*, not

merely input strings. A test instance is OOD when its signature is absent from both the SFT and RL supports, even though its constituent atomic skills and routers may have appeared elsewhere.

**SFT training.** SFT uses only filtered correct reasoning traces. For each training prompt $P_{\sigma,x}$, the model is supervised to reproduce one of the retained reasoning-inclusive responses ending in the correct final JSON answer. The function definitions are not included in the SFT prompt; therefore, the model must rely on the function identifiers and the compositional expression rather than in-context access to the implementation. Importantly, the latent signature $\sigma$ is used only by the data generator and verifier. The SFT objective does not directly label individual skill variables $i_t$ or routing variables $\rho_t$.

**RL training.** RL starts from the SFT checkpoint and uses the same prompt format, again with function definitions hidden. For each prompt, the current policy generates a group of 16 rollouts. The verifier parses the final JSON answer and compares it with the deterministic gold output $Y(\sigma, x)$. The reward is therefore a final-answer exact-match signal only. No intermediate values, routers, skill indices, or gold reasoning traces are provided to the RL objective. The on-policy rollouts nevertheless expose the model to multiple trajectories for the same prompt, and the correctness signal reinforces trajectories that compose the latent skills and routers successfully.

**How interventions are implemented.** All controlled interventions are implemented by modifying the signature supports before sampling input strings. For skill-removal experiments, we remove from $\mathcal{C}_{\mathrm{RL}}$ every signature containing a held-out skill $f_\star$. For router-removal experiments, we analogously remove every signature containing the held-out router $\rho_\star$. Re-injection is performed by adding back either isolated signatures, such as $L = 1$ uses of $f_\star$, or compound signatures in which the held-out atom appears together with other skills and routers. For SFT–RL distribution experiments, we construct $\mathcal{C}_{\mathrm{SFT}}$ and $\mathcal{C}_{\mathrm{RL}}$ to have the prescribed subset, superset, overlap, or disjoint relationship, while keeping the atomic inventory and optimization hyperparameters fixed.

**Why this DGP is tied to the identifiability claims.** The experiment separates three objects that are conflated in natural reasoning data. First, the prompt descriptor $P_{\sigma,x}$ specifies the visible composition expression and input string. Second, the latent signature $\sigma$ specifies the unobserved sequence of skill and routing selections. Third, the observed trace $D$ is the model-generated text. Because $\sigma$ is known to the generator, we can control whether training provides coverage of individual atoms, compound interfaces, and OOD recombinations. Because $\sigma$ is not provided as a training label during RL, improvements on held-out signatures cannot

be explained by direct supervision of the latent variables; they require the model to recover reusable skill and routing behavior from traces and outcome rewards.

## J  Longer-Run Convergence Results

We continue the experiments from Figures 5 and 7 to 1000 RL training steps. Figures 14 and 15 show that the corresponding trends remain stable over longer training.

## K  A Concrete Algebra Example

In Figure 16, we provide a concrete algebra example to illustrate skills, routing mechanisms, and identification conditions (Figure 2, Condition C.1).

## L  Representation-Level Experiment with SAE

We applied pretrained SAEs (Gao et al., 2025) to Gemma-2-2b-it and run SAE-based causal discovery on real-world math benchmarks. The results are presented in Figure 17. The mapping follows a principled criterion grounded in our theory: nodes whose activation patterns correspond to specific mathematical operations are classified as skill nodes – they perform atomic computational steps. Nodes that govern which operation to apply given the current problem state (e.g., selecting which intermediate result to carry forward) are classified as routing nodes – they control the composition structure. Crucially, this hierarchy is discovered from learned representations via causal discovery, not imposed a priori, confirming the theory's prediction that the skill-routing structure emerges in real model internals.

## M  Sensitivity Tests over RL Hyperparameters

We complete the full sensitivity experiments (shown in Figure 18), sweeping number of samples per prompt, sampling temperature, reward shaping variants, and KL/entropy regularization coefficients. Across all configurations, the findings remain consistent, confirming that our results are robust to RL algorithmic details.

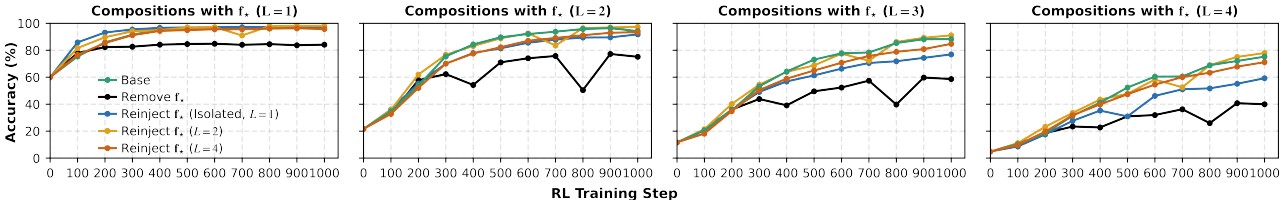

*Figure 14.* **Longer-run version of Figure 5.** As training proceeds to 1000 RL steps, the curves stabilize and the relative ordering remains unchanged.

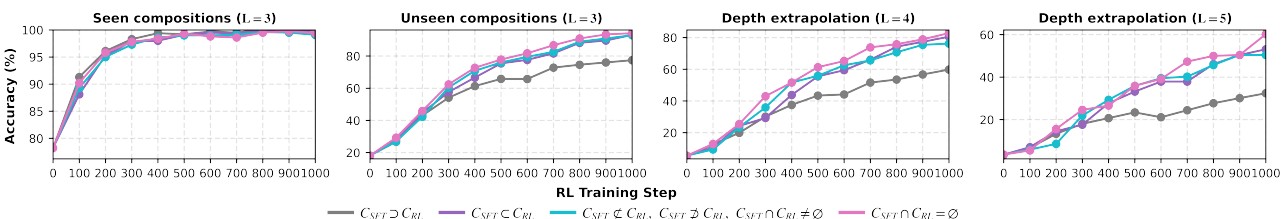

*Figure 15.* **Longer-run version of Figure 7.** The ordering between distribution-overlap regimes persists as training proceeds to 1000 RL steps.

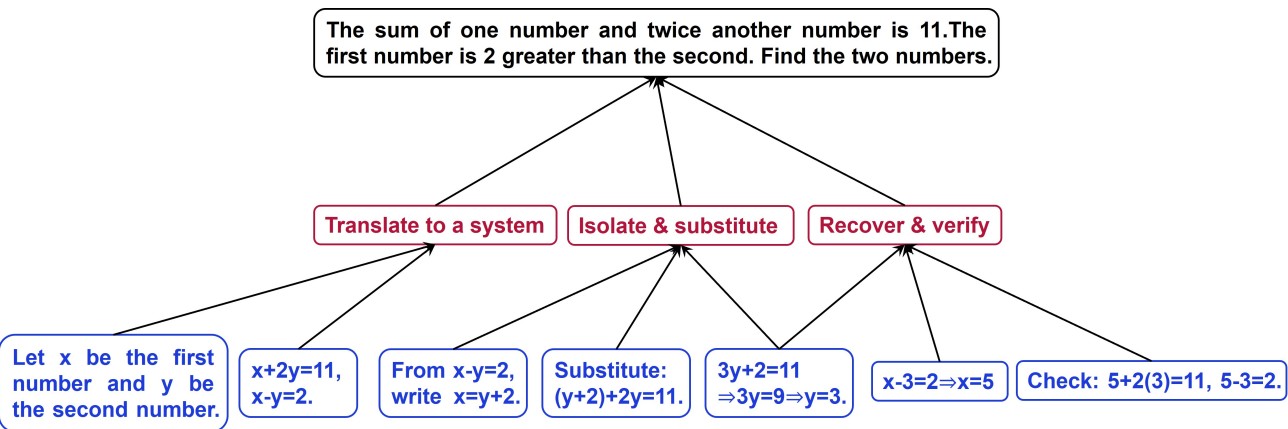

*Figure 16.* **A concrete algebra example illustrating skills, routing mechanisms, and identification conditions (Figure 2, Condition C.1).** The **black box** is the problem descriptor $\mathbf{P}$; the **red boxes** are latent selection variables; the **blue boxes** are observed trace tokens. **Neighborhood coverage** (Condition C.1-v): If "Translate to a system" only ever preceded "Isolate & substitute," it could be mistaken for one fused routine; seeing it also before "Eliminate by subtraction" ( $(x + 2y) - (x - y) = 11 - 2 \Rightarrow 3y = 9 \Rightarrow y = 3$ ) provides a distinct neighborhood that isolates it as reusable. **Local distinguishability** (Condition C.1-v,vi): "Substitute: $(y + 2) + 2y = 11$" depends only on "Isolate & substitute" (no other red box), serving as its *anchor*; choosing elimination instead would produce a visibly different token, so distinct module states remain separable. **Restricted choice sets** (Condition C.1-iii): After "Substitute: $(y + 2) + 2y = 11$," the algebraically valid continuation is "$3y + 2 = 11 \Rightarrow 3y = 9 \Rightarrow y = 3$"; other token sequences would violate the problem's constraints, so valid traces form a strict subset of all combinatorial possibilities.

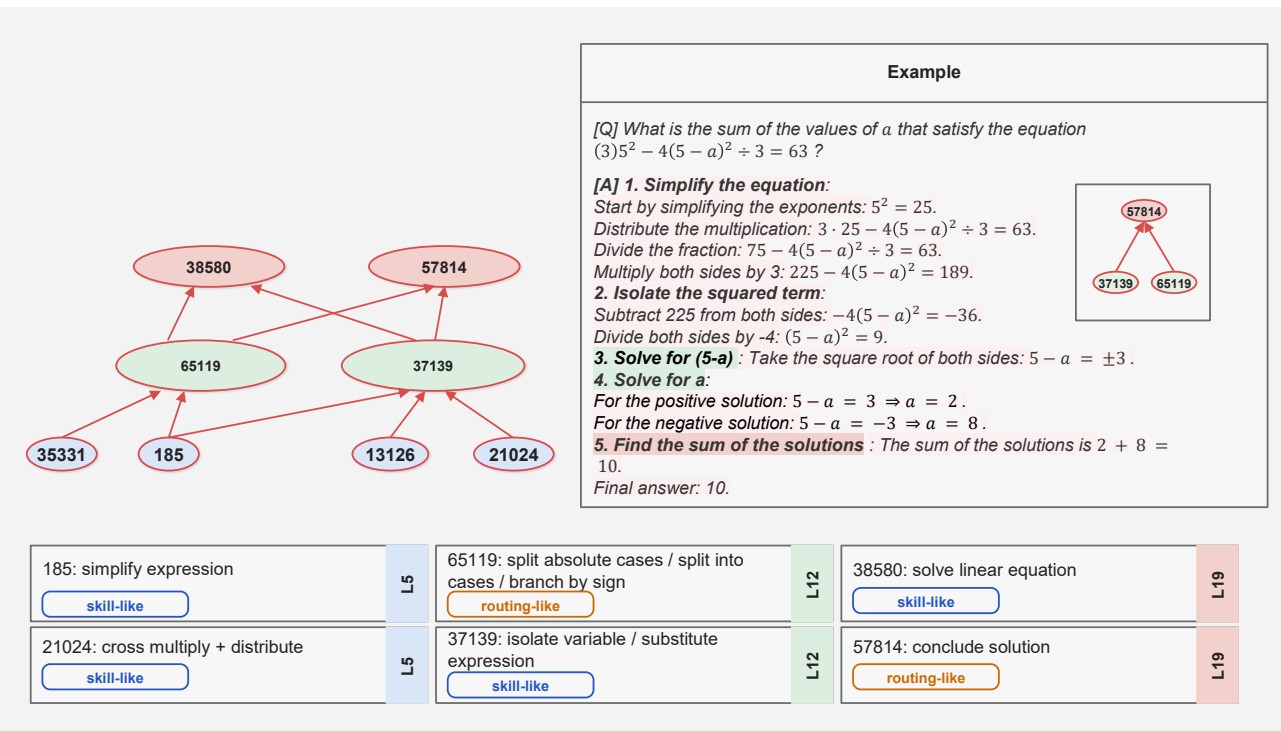

*Figure 17.* **SAE-based causal graph on mathematical reasoning traces.** Based on real-world math benchmarks, we apply pretrained SAEs to Gemma-2-2b-it, partition tokens by sequence position, and run causal discovery across SAE features. The graph contains skill-like nodes corresponding to mathematical operations and routing-like nodes corresponding to choices about which operation to apply given the current problem state.

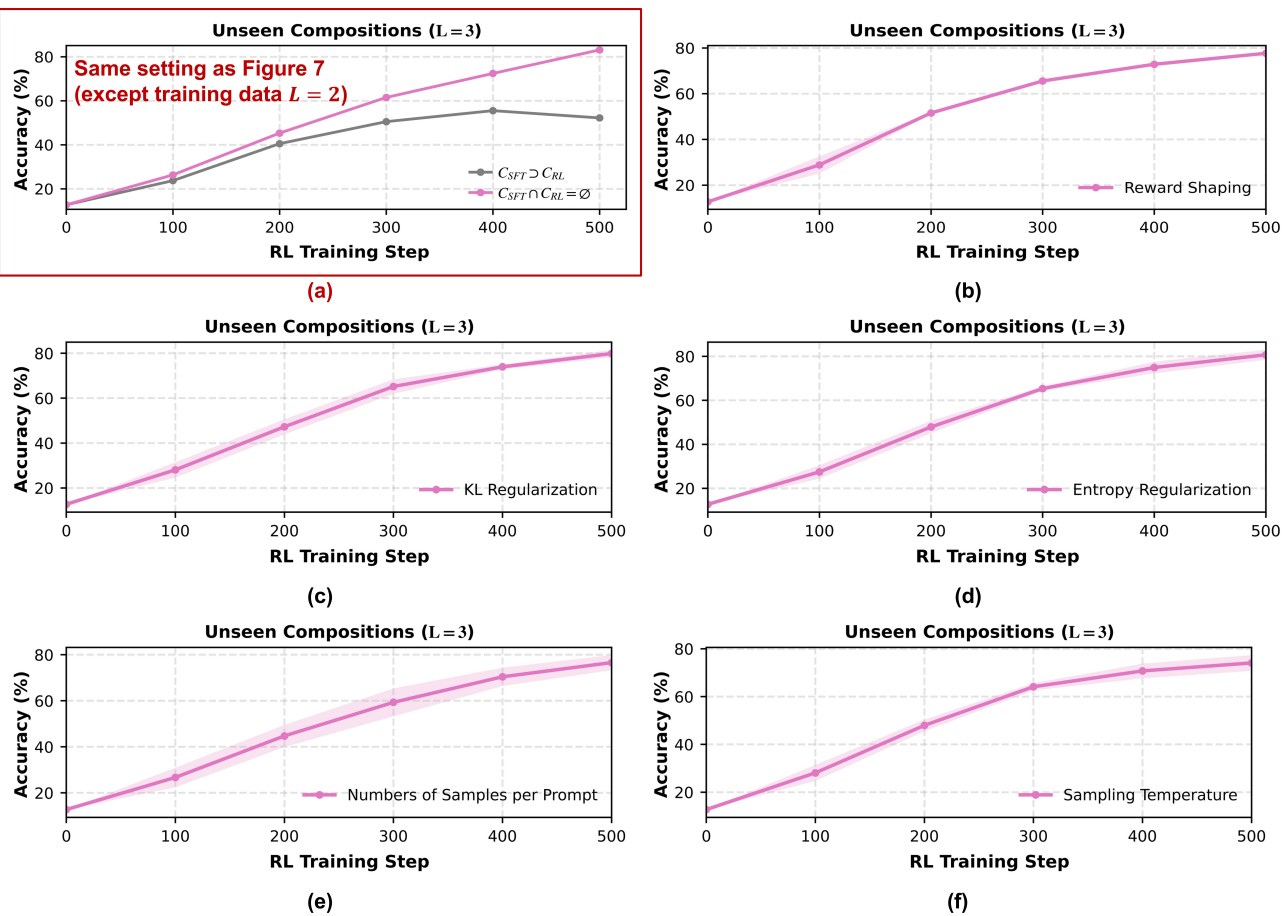

*Figure 18.* **Full sensitivity experiments.** Panel (a) follows the setting of Figure 7, except that the training data use level $L = 2$ rather than $L = 3$. Panels (b)-(f) vary overlong reward shaping (true or false), KL regularization (coefficient 0, 1e-4, 1e-3), entropy regularization (coefficient 0, 5e-4, 1e-3, 2e-3), number of samples per prompt (8, 14, 16), and sampling temperature (0.5, 0.7, 1.0). In each sensitivity experiment, the solid line reports the average over hyperparameter values, and the shaded region shows variation across these runs. The qualitative conclusions remain stable across these RL algorithmic details.

