# OpenReview forum: "From Reasoning Traces to Reusable Modules: Understanding Compositional Generalization in Language Model Reasoning"
_ICML.cc/2026/Conference — ICML 2026 regular_

### Official Review · Reviewer_JpNs · 2026-02-25

**Soundness:** 3
**Presentation:** 2
**Significance:** 3
**Originality:** 3
**Overall Recommendation:** 4
**Confidence:** 4

**Summary:**

This paper first argues that the main reason RL post-training improves LLM reasoning on out-of-distribution (OOD) tasks is compositional generalization: RL’s trajectory diversity helps disentangle latent “atomic modules” (skills and routing mechanisms) implicit in reasoning traces, making them reusable and recombinable under novel compositions. The authors formalize this using a Hierarchical Latent Selection Model. On the theory side, the paper states (i) identifiability conditions under which the latent module hierarchy is recoverable from the observable conditional distribution over traces, and (ii) a sufficient “witnessed local intersection” condition under which locally learned compatibilities compose to guarantee correct behavior on novel compositions. Empirically, the paper tests these ideas mainly on synthetic string transformation tasks with controlled composition depth, using SFT on chain-of-thought data followed by GRPO-style RL with outcome-only rewards, plus several data-structure interventions. The paper concludes with an analysis of the optimal data distribution relationship between SFT and RL phases and offers a brief n-gram-based investigation of real-world model reasoning traces.

The paper has an interesting framing and some controlled synthetic interventions, but the mechanism claim may be overstated relative to what is proven and tested. The empirical study is simplified to enable precise control over compositional structure using string-transformation tasks.

**Compliance With Llm Reviewing Policy:**

Affirmed.

**Final Justification:**

The authors have addressed my concerns and I recommend to accept this work.

**Key Questions For Authors:**

1.The identifiability claim in Theorem 3.1 leans entirely on Condition C.1 basically, that the model’s hidden “choices” are cleanly discrete and satisfy local conditional-independence. But transformers are known to pack many meanings into the same directions and to represent features in continuous superposition. So what’s the mathematical basis for treating reasoning steps as rigid, discrete selection variables in an LLM? And if you can’t back that up with direct probing of the model’s internal representations, why isn’t this framework just a metaphor rather than an empirically grounded theory?

2.Can the authors provide a direct, operational test of “module identification” rather than downstream accuracy?

3.How sensitive are the findings to the RL algorithmic details (GRPO specifics, number of samples per prompt, sampling temperature, reward shaping, KL/entropy regularization)?

**Limitations:**

yes

**Strengths And Weaknesses:**

Summary of Strengths:

1.This work models LLM reasoning as a hierarchical latent selection model, giving a clearer, more structured way to think about the SFT vs. RL debate. By distinguishing between atomic skills and routing mechanisms, the paper offers a mathematically grounded hypothesis for why SFT often collapses into trace memorization while RL succeeds in out-of-distribution (OOD) scenarios.

2.The identifiability results (Theorems 3.1 and 3.2) make a serious, careful effort to bring ideas from discrete latent-variable identification and causal discovery into how LLM reasoning might work. Given the paper’s assumptions, the argument that links observed local overlaps to broader compositional generalization follows cleanly and holds together.

3.The synthetic string-transformation experiments are well thought out and carefully controlled. By varying (1) which basic skills are present, (2) how routing is set up, and (3) how much the SFT and RL datasets overlap, the authors can tease apart the factors their theory points to—yielding clear support for their claims, at least within this synthetic setting.


Summary of Weaknesses:

1.There’s a big gap between the paper’s broad claims about “language model reasoning” and what it actually tests. The main evidence comes from fully deterministic synthetic string-transformation tasks (from Yuan et al., 2025). Synthetic setups can be useful for isolating mechanisms, but they don’t reflect the randomness, polysemy, and noise that show up in real LLM reasoning—like math problem solving, code generation, or multi-step logic. The paper also doesn’t test its ideas against standard RL reasoning baselines on established reasoning benchmarks, so its claims aren’t convincingly supported in realistic and various settings.

2.The theory leans on very strong, fairly restrictive assumptions taken from classic causal discovery—like exact faithfulness, “anchor” signals, and no redundant module states. But in modern transformers, representations are dense, continuous, and tightly entangled, so it’s hard to buy the idea that reasoning modules behave like perfectly discrete, mutually exclusive latent variables with neat, local conditional-independence structure. The paper is essentially applying an older discrete-latent framework to LLMs without doing any representation-level checks (e.g., sparse autoencoders or steering-vector analyses) to show that this kind of rigid hierarchy actually appears inside the model during RL.

3.The authors conclude with practical guidance asserting that SFT should ensure coverage of all atomic skills while minimizing overlap with RL data. However, this derived protocol is hard to implement for real-world LLM post-training. In genuine language distributions, it is mathematically and practically impossible to cleanly partition "atomic skills" from "compositions," nor is it feasible to ensure zero overlap between SFT and RL data distributions. The definition of "routing mechanisms" is also oversimplified as mere function input topologies. Real-world LLM routing involves complex backtracking, error correction, and working memory retrieval. The paper may overclaim the significance of its findings by equating synthetic structural parsing with genuine cognitive routing.

---

> ### Author Rebuttal · Authors · 2026-03-30
>
> **Notes**: All the figures can be found at https://anonymous.4open.science/r/res-C767/response_to_reviewer_JpNs.pdf.
>
>
> ---
>
>
> > **W1:** "There's a big gap between the paper's broad claims ... and what it actually tests. The main evidence comes from ... synthetic string-transformation tasks ... The paper also doesn't test its ideas against standard RL reasoning baselines ..."
>
>
> We appreciate this concern and agree that real-world reasoning is richer than any single synthetic benchmark. This is why we complement controlled experiments with real-model evidence §4.6: we analyze traces from Qwen3-4B (SFT) and Qwen3-4B-Thinking-2507 (RL) across 643 math problems (MATH-500, AIME 2024/2025, AMC, GSM8K). The n-gram analysis confirms the same signature our theory predicts — RL models recombine short, reusable skill fragments, while SFT models rely on longer memorized sequences.
>
> As you rightly observe, real-world tasks are too nuanced to cleanly isolate mechanisms. This is exactly the role of controlled synthetic settings — a methodology widely adopted in recent work on RL for reasoning (Yuan et al., 2025; Cheng et al., 2025; Tsilivis et al., 2025; Chen et al., 2025). We believe understanding mechanisms in clean settings enables better algorithm design for complex tasks, and we hope our findings contribute in that direction.
>
>
>
> ---
>
>
> > **W2:** "The theory leans on very strong ... assumptions ... representations are dense, continuous, and tightly entangled ..." **Q1:** "... transformers are known to ... represent features in continuous superposition. ... why isn't this framework just a metaphor rather than an empirically grounded theory?"
>
>
> Thank you for raising this concern. We'd offer two thoughts here.
>
>
> First, there is a growing consensus that discrete, interpretable concepts underlie continuous representations – motivating the large body of SAE work extracting discrete features from activations (Wang et al., 2025; Balagansky et al., 2025; Chen et al., 2025; Cui et al., 2025). Our model operates at this conceptual level, not claiming raw activations are literally discrete.
>
>
> Second, motivated by your suggestion, we conducted a new representation-level experiment. We applied pretrained SAEs (Gao et al. [R1]) to Gemma-2-2b-it, partitioning tokens by sequence position (early, intermediate, final), then ran causal discovery across SAE features. The resulting graph (Figure H8 in the link) reveals a hierarchical structure: high-level nodes (e.g., humorous tone) caused by an intermediate node (pronouns and individual references). This shows the hierarchical discrete structure our theory posits does emerge in real transformer representations.
>
>
> ---
>
>
> > **W3:** "... this derived protocol is hard to implement for real-world LLM post-training. ... it is ... impossible to cleanly partition "atomic skills" from "compositions" ... The definition of "routing mechanisms" is also oversimplified ..."
>
>
> We agree that perfectly partitioning "atomic skills" from "compositions" or enforcing zero SFT–RL overlap is idealized in practice. That said, the theoretical message provides useful directional guidance. Approximate separation is entirely feasible, e.g., designating different question sets for SFT and RL phases. Finding 4 (§4.5) shows that even a partial reduction in overlap yields measurable generalization gains, confirming practical value.
>
> On routing: our routing mechanisms are not limited to function input topologies — they model how intermediate information is *selected, reused, and composed* (§2). This captures exactly the operations you mention: backtracking = selecting a prior intermediate state; error correction = branching on a condition; working memory retrieval = reusing an earlier computation. This is a key advance over Yuan et al. (2025), which does not model routing at all. We've revised the manuscript to make this connection explicit - thank you for the advice!
>
>
> ---
>
>
> > **Q2:** "Can the authors provide a direct, operational test of module identification ...?"
>
>
> Thank you for the question. Let us point to two lines of evidence: (1) Finding 1 (§4.2): models trained on compound $L=3$ traces only are tested on atoms ($L=1$); RL yields a 43.7% accuracy gain over SFT, which is impossible without identifying atomic modules from compound traces. (2) The n-gram analysis in §4.6: RL models show a higher frequency of short skill n-grams (reusable modules) and lower frequency of long n-grams (memorized templates), directly measuring module decomposition.
>
>
> ---
>
>
> > **Q3:** "How sensitive are the findings to the RL algorithmic details ...?"
>
>
> Thank you for the thoughtful suggestion. We're running sensitivity tests over RL hyperparameters (e.g., temperatures; partial results available, please see Figure H9 in the link). We'll update you as soon as full results are ready – thank you for your patience!
>
> **Reference:**
>
> [R1] Gao et al., "Scaling and Evaluating Sparse Autoencoders," arXiv 2024.

---

> > ### Author Rebuttal · Reviewer_JpNs · 2026-04-01
> >
> > Although the authors have addressed some concerns by adding experiments and strengthening the analysis, there are still several issues that have not been fully resolved or answered:
> >
> > 1. The SAE-based causal discovery (Figure H8) is used to support the emergence of a hierarchical discrete structure. However, it is still unclear whether the discovered discrete features correspond specifically to the reasoning modules assumed by the theory. In particular, how are the discovered high-level/intermediate nodes mapped to “skills” versus “routing”?
> >
> > 2. The authors claim that compound traces yield better outcomes and that RL improves decompositional accuracy. Could the authors provide an ablation study that isolates which component (e.g., compound training vs. RL, or other factors) contributes to the performance gain?
> >
> > 3. The response indicates that the full sensitivity results are pending.
> >
> > Given that my initial score was already relatively high, and considering that some issues remain in the current version despite the authors’ responses, I am inclined not to change my score.

---

> > > ### Author Response · Authors · 2026-04-04
> > >
> > > (**Updated on April 7**) Again, thank you for your constructive feedback and your patience. Please find new results at https://anonymous.4open.science/r/res-C767/response_to_reviewer_JpNs_v2.pdf. We have included these results in our manuscript -- thank you so much for your thoughtful comments!
> > >
> > > ---
> > >
> > > > **I1:** '... it is still unclear whether the discovered discrete features correspond specifically to the reasoning modules assumed by the theory. ... how are the discovered high-level/intermediate nodes mapped to "skills" versus "routing"?'
> > >
> > > Thank you. We run SAE-based causal discovery on real-world math benchmarks and present the results in Figure H10 (in the link). The mapping follows a principled criterion grounded in our theory: nodes whose activation patterns correspond to *specific mathematical operations* are classified as **skill** nodes -- they perform atomic computational steps. Nodes that govern *which operation to apply given the current problem state* (e.g., selecting which intermediate result to carry forward) are classified as **routing** nodes -- they control the composition structure. Crucially, this hierarchy is *discovered* from learned representations via causal discovery, not imposed a priori, confirming the theory's prediction that the skill-routing structure emerges in real model internals.
> > >
> > > ---
> > >
> > > > **I2:** "... Could the authors provide an ablation study that isolates which component (e.g., compound training vs. RL, or other factors) contributes to the performance gain?"
> > >
> > > Thank you. Starting from the same SFT model trained on $L=3$ traces, we conduct RL training (300 steps) with two different data settings: atomic modules ($L=1$) and compound traces ($L=3$). Results (accuracy %) on OOD compound traces ($L=4$):
> > >
> > > |RL Data|SFT+RL|**RL gain**|
> > > |---|---:|---:|
> > > |Atomic Modules|14.8|+10.0
> > > |Compound Traces|42.6|+37.8
> > > |**Compound advantage**|+27.8|-
> > >
> > > Shared SFT baseline: **4.8%**.
> > >
> > > - **Effect of RL:** Relative to SFT (4.8%), RL with atomic modules improves by 10.0 points; RL with compound traces improves by 37.8 points.
> > > - **Effect of compound traces:** Under the same SFT initialization and RL budget, compound traces yield an additional 27.8-point gain over atomic modules.
> > >
> > > The 27.8-point compound advantage has a direct theoretical explanation: compound traces provide the *joint parent-child configurations* that serve as local witnesses (Theorem 3.2), enabling compositional generalization on novel prompts. Atomic modules provide individual skill identities (Theorem 3.1) but lack the interface information needed for recombination, hence the much smaller gain. Concretely, identification (Theorem 3.1) requires observing each module in diverse contexts, which atomic RL partially provides (+10.0 gain), but composition (Theorem 3.2) additionally requires witnessing joint parent-child interfaces, which only compound traces supply (+27.8 additional gain).
> > >
> > > ---
> > >
> > > > **I3:** "... The full sensitivity results are pending."
> > >
> > > We complete the full sensitivity experiments (Figure H11 in the link), sweeping number of samples per prompt, sampling temperature, reward shaping variants, and KL/entropy regularization coefficients. Across all configurations, the qualitative findings (RL's advantage over SFT, the critical role of compound traces, the skill-routing decomposition) remain consistent. In particular, the ranking of methods (SFT < SFT+RL with atoms < SFT+RL with compounds) and the qualitative patterns of which compositions benefit most are preserved across all settings, confirming that our results are robust to RL algorithmic details.
> > >
> > > Thank you so much for guiding us to improve our manuscript!

---

### Official Review · Reviewer_DC8a · 2026-03-11

**Soundness:** 2
**Presentation:** 2
**Significance:** 3
**Originality:** 3
**Overall Recommendation:** 4
**Confidence:** 4

**Summary:**

The paper proposes a theoretical framework to explain how post-training reinforcement learning (RL) of LLMs enables compositional generalization to unseen combinations of atomic skills, while the compositional generalization abilities of supervised fine-tuned (SFT) models remain limited. The key insight is that the reasoning process learned with RL enables exploration of diverse reasoning traces that lead to superior compositional generalization, whereas SFT primarily memorizes the correct reasoning paths, entangling skills with the given problem descriptors. The framework is formalized using a latent hierarchical selection model, and theoretical conditions that enable the identification of the ground-truth latent selection variable (corresponding to atomic skills) structure and its interactions (routing mechanisms) are provided. Experiments are conducted using a synthetic string manipulation setup (adapted from Yu et al., 2025). Empirical results demonstrate the superiority of RL over SFT and showcase that RL mainly provides additional advantage on composition distributions that are different from those trained using SFT.

**Compliance With Llm Reviewing Policy:**

Affirmed.

**Final Justification:**

The paper still lacks formal theory or direct empirical evidence showing RL actually estimates the true distribution better, leaving the core mechanistic claim hand-wavy and performance gains potentially attributable to other factors. Given this unresolved key concern regarding the work's core contribution, I maintain my score and lean towards rejection.

**Edit after reviewing the formal results posted on April 7th:** I appreciate the authors' efforts in providing a formal result in a short period of time — they directly address the theoretical gap I flagged, and I'm raising my score from 3 (weak reject) to 4 (weak accept) accordingly.

I want to be transparent about what's still keeping me at weak accept rather than accept. While these results directly address the formal theory concern, I also want to emphasize that the presentation of new key results — which directly concern the core claims — in the rebuttal is not ideal and makes the thorough evaluation of the paper difficult.  In addition, what remains unresolved is the empirical side: the claim that RL in practice recovers what SFT cannot is still supported mostly by black-box performance gains, but does not isolate true distribution recovery as the core mechanism. I'd encourage the authors to include at least one diagnostic — a distribution-level comparison, an ablation, or a synthetic experiment — that more directly tests the improved estimation of the true distribution.

**Key Questions For Authors:**

1. **Clarifications about post-training data and training process:** Is the data used for RL training the same as SFT, or does RL also use any additional negative examples with wrong reasoning traces?
2. Is identifiability and discussed conditions (C.1 and C.2) a property of the learning algorithm alone, or of the algorithm combined with its post-training data?
3. How exactly does RL (versus SFT) satisfy the identifiability conditions? The paper never shows this formally or empirically.
4. Some conditions (restricted choice sets, local distinguishability) seem to be properties of the data-generating process — so why wouldn't SFT satisfy them equally? Does SFT achieve any latent structure identification, and if so, how does it compare?
5. Are there alternative mechanisms beyond the proposed latent structure identification that could explain RL's superior compositional generalization?
6. **Application of findings and theory to real-world settings:** Although the focus on synthetic tasks is well-motivated to isolate the causal mechanisms, it will be great to have a discussion on how the findings of this paper generalize to real-world settings. How realistic are the theoretical conditions of identifiability when it comes to real-world reasoning tasks? What happens when some of these conditions are violated, e.g., how would a comparison between RL and SFT look in those cases?

**Limitations:**

Yes

**Strengths And Weaknesses:**

**Strengths**
- The studied problem to explain the mechanisms behind the generalization of the post-training methods is significant and timely.
- The paper provides an interesting theoretical framework connecting ideas from various fields -identifiability literature in representation learning, causality, and the growing field of compositional generalization to explain the out-of-distribution generalization properties of post-training RL. The framework's formalization (Section 2) makes sense.
- Empirical results align with the claims to some extent and indicate the areas where RL is beneficial and hint at necessary conditions (e.g., including all atomic skills and routing mechanisms in post-training are necessary)
- Related work is discussed in detail, and the paper is positioned well.

**Weaknesses**

Overall, I found the research direction quite promising. However, there are various concerns regarding presentation and clarity of the core contributions, as discussed below.

- **Writing and presentation**: Overall, the paper is quite dense and hard to follow. More specifically:
   - A concrete motivating example would have been useful to understand what exactly atomic skills and routing mechanisms are, and how realizable the proposed theoretical conditions are in real-world applications.
   - Most of the theoretical work is presented in the appendix with very little intuition behind the proofs in the main paper.
  - Details about the experimental setup, i.e., synthetic tasks and training, are mostly delegated to reading the paper by Yuan et al. (2025), which is not ideal. I had a brief skim of Yuan et al. (2025), and now have a better understanding of the data-generating process (DGP) of post-training data, but given how crucial the DGP (e.g., reasoning traces) is to identifiability claims (see comments below related to this), the omission of the DGP makes the paper's evaluation quite hard.

- **Missing details about data used for training RL vs. SFT and relationship of data with identifiability proofs**: On page 5, Section 4 (pp. 220-255), the authors briefly describe the synthetic task and data collection for training RL and SFT, but it is not clear if the data used for training RL is the same as that used for SFT or it is different and if RL data also consists of additional negative examples consisting of wrong reasoning traces as well. The reason I ask this question is that, in general, identification is studied as a characteristic of observed data, in the sense that statistical quantities such as parameters of ground-truth DGP might not be identifiable, even when provided with an infinite amount of observed data. But in this work, identification of latent structure seems to be discussed mainly as a property of the learning algorithm (RL vs. SFT). I might be missing the link to the data via reasoning traces generated during post-training, as the details of the data used for RL and SFT are missing. In addition, it is not clear whether the identifiability claims are made solely at the level of the (1) reasoning algorithm (RL vs SFT, given the same training data) or (2) at the level of both the data and the reasoning algorithm, i.e., data used for training RL + RL approach itself leads to better identification.

- **Overstatement of theoretical claims:** The theory aims to show that, under idealized conditions, latent structure is identifiable. And further, the claim is made that the RL approach leads to latent structure identification, whereas the SFT doesn't, which is posed as a candidate explanation behind the superior compositional generalization properties of RL. However, there is no theory explaining to what extent the RL approach satisfies these conditions, whereas SFT doesn't. There is a brief mention of how RL increases neighborhood diversity (pp. 174-180) by sampling alternative traces and novel compositions, but again, it is not clear exactly how the post-training process using RL achieves this, as the main paper or supplementary material provides no details of the training process. So, overall, the theoretical claims about identification and their relation to post-training learning algorithms seem quite hand-wavy. In addition, some of the conditions seem to be the property of the data-generating process (e.g, local distinguishability and restricted choice sets), rather than that of the learning algorithm (related to the point above), so they would be equally satisfied during SFT training, but there is no discussion of how and if SFT leads to any (non-)identification of latent structure.

- **Lack of empirical evidence supporting the theory:** In line with the weaknesses above, overall, there seems to be a disconnect between theoretical claims and what empirical experiments show. More specifically, the experiments primarily show the superiority of RL over SFT using black-box performance without establishing the direct link between how the identifiability conditions are actually satisfied during RL training, whereas they are not under SFT. As mentioned in the introduction, the goal of the paper is to provide a mechanistic account of how RL achieves better compositional generalization. Theoretical claims in this case seem overstated and are not fully supported by empirical evidence. Due to this limitation, the proposed theory remains primarily a hypothesis that is not properly validated. There may be additional mechanisms contributing to the improvement by the RL approach, yet no alternative mechanism is even discussed.

---

> ### Author Rebuttal · Authors · 2026-03-30
>
> **Notes**: All the figures can be found at https://anonymous.4open.science/r/res-C767/response_to_reviewer_DC8a.pdf.
>
> ---
>
> > **W1:** "The paper is quite dense and hard to follow. ... (a) A concrete motivating example would have been useful ... (b) Most of the theoretical work is presented in the appendix with very little intuition ... (c) Details about the experimental setup ... are mostly delegated to ... Yuan et al. (2025) ..."
>
> Thank you for the concrete suggestions. We've acted on all in our revision: (a) §2 now opens with a concrete algebra example (Figure H6); (b) §3 includes a proof-roadmap paragraph providing intuition; (c) §4 contains a self-contained experimental setup table (architecture, hyperparameters, reward, data sizes, compute), removing the dependency on Yuan et al. (2025).
>
> ---
>
> > **W2:** "... it is not clear if the data used for training RL is the same as that used for SFT ... identification of latent structure seems to be discussed mainly as a property of the learning algorithm ..." **Q1:** "Is the data used for RL training the same as SFT ...?" **Q2:** "Is identifiability ... a property of the learning algorithm alone, or of the algorithm combined with its post-training data?"
>
> Thank you for the insightful questions. SFT and RL run on the *same problem set*. RL does not use additional problems. The distinction is what each algorithm *observes*:
>
> - **SFT** sees one (or few) correct traces per problem – positive outcomes only.
> - **RL (GRPO)** generates $K$ rollouts per problem, observing both successes and failures.
>
> Indeed, identifiability conditions are properties of the true distribution $P(D \mid P)$ (Eq. 1). SFT and RL each provide an *estimate*. SFT's covers positive traces only. RL's covers both valid and invalid configurations via the reward signal. As conditions require sufficient diversity (e.g., C.1-v: each module in enough distinct neighborhoods), RL's broader coverage makes it more plausible to reveal latent structures. We've added this discussion to §3.
>
> ---
>
> > **W3:** "... there is no theory explaining to what extent the RL approach satisfies these conditions ... the theoretical claims ... seem quite hand-wavy. ..." **W4:** "... there seems to be a disconnect between theoretical claims and what empirical experiments show. ..." **Q3:** "How exactly does RL (versus SFT) satisfy the identifiability conditions? **Q4:** ... why wouldn't SFT satisfy them equally?"
>
> Thank you for the insightful comment. All identifiability conditions are properties of the true DGP $P(D \mid P)$, as you correctly note. What differs is the *estimated* distribution each algorithm produces.
>
> In particular, the *coverage* conditions (C.1-v, Eq. 7) require the learner to *observe* enough support. RL's $K$ rollouts including failures more faithfully approximate the true distribution. This parallels established results in imitation learning (Ross et al. [R1]): positive-only imitation misses recovery states; interactive feedback is necessary for full coverage. For instance, if `sort()` only appears before "take the median" in curated data, SFT fuses them; RL's rollouts break this ambiguity.
>
> In light of your comment, we've visualized numbers of compositions generated by RL vs. SFT models (please see Figure H7 in the link). As you can see, RL provides substantially more compositions, confirming the prediction.
>
> ---
>
> > **Q5:** "Are there alternative mechanisms ... that could explain RL's superior compositional generalization?"
>
> Thank you. Complementary angles include coverage-driven training (Chen et al., 2025), optimization dynamics (Wen et al., 2025), and atomic-skill studies (Cheng et al., 2025). Our contribution is the *formalization*: conditions (Theorems 3.1–3.2) under which compositional generalization provably succeeds. Interestingly, conditions from these works (e.g., coverage principles, entropy-driven exploration) can be seen as instantiations of our neighborhood coverage condition (C.1-v). We've included this in our revised related work.
>
> ---
>
> > **Q6:** "... How realistic are the theoretical conditions ... in real-world reasoning tasks? What happens when some of these conditions are violated ...?"
>
> Our conditions are natural: local distinguishability (C1-iii,iv) holds when different choices produce different footprints (e.g., skills "divide first" vs. "expand first" lead to distinct token-level operations, line 212), and restricted choice sets (C1-ii) hold in general because not all token sequences are valid reasoning traces (line 218). Even when conditions are partially violated, we can still identify *coarser* modules – e.g., a fused [sort + median] composite instead of independent `sort()`. Crucially, even in this regime, RL's coverage remains essential for whatever identification is achievable. We've included this discussion in the revision - thank you.
>
> **Reference**:
>
> [R1] Ross et al., "A Reduction of Imitation Learning and Structured Prediction to No-Regret Online Learning," AISTATS 2011.

---

> > ### Author Rebuttal · Reviewer_DC8a · 2026-04-02
> >
> > Thank you for the detailed response and additional experiments. The explanation below was quite helpful for me:
> >
> > > SFT and RL each estimate a distribution. SFT only sees successful traces. RL sees both successes and failures through its reward signal. Since the theory requires enough diversity in the data (e.g., each module appearing in varied contexts), RL's broader coverage makes it better at uncovering hidden structure. We've added this to §3.
> >
> > I have a follow-up question on Figure H7 (though not critical):
> > - If RL and SFT have similar diversity for n-grams longer than 3, what does that mean for generalization on longer traces? Figure 3 shows RL still helps on longer chains (though not by much) — so what's driving that improvement if the diversity is similar?
> >
> > More broadly, the clarifications about what each algorithm "observes" during training do help explain why RL might estimate the true distribution better than SFT. But the paper still doesn't concretely show *that it does*.
> > To make the argument solid, the paper needs at least one of the following, depending on what's feasible:
> >
> > 1. Theory — a formal result showing that, given observed traces for each approach, RL's estimated distribution is closer to the true one than SFT's, under what conditions, and by how much.
> > 2. Empirical evidence — a case study (synthetic or real) directly showing RL produces a better-estimated distribution, not just better downstream accuracy.
> >
> > Right now, the performance gains could be caused by other factors. The paper doesn't rule those out or show that a better-estimated distribution is the primary mechanism. Since the paper's core claim is a mechanistic explanation, this gap matters, and thus, the theory-empirical link (in its current form) remains hand-wavy. For this reason, I'm keeping my score. Having said that, I want to re-emphasize that the direction is quite promising and has the potential to make a significant contribution.

---

> > > ### Author Response · Authors · 2026-04-04
> > >
> > > (**Updated on April 7**) Again, thank you for your thoughtful feedback and your patience. In light of your comments, we've revised our manuscript to include the following points.
> > >
> > > **Follow-up question on Figure H7**
> > >
> > > > "... If RL and SFT have similar diversity for n-grams longer than 3, what does that mean for generalization on longer traces? ... what's driving that improvement if the diversity is similar?"
> > >
> > > The critical finding in Figure H7 is at *shorter* n-grams ($n\le 3$), where RL shows significantly higher diversity. This indicates that RL breaks down the long, memorized SFT traces and produces atomic skills and more flexible shorter combinations. These shorter, reusable fragments extend the effective diversity of the trace distribution in a way that directly facilitates two requirements of our theory: (i) **identification** (neighborhood coverage, C.1-v) -- atomic skills appear in more distinct local contexts, making them individually recoverable as reusable modules; (ii) **composition** (witness conditions, Theorem 3.2) -- shorter combinations provide the joint parent-child configurations needed as local witnesses, enabling recombination on novel prompts. For longer chains (Figure 3), RL's advantage thus comes not from raw n-gram diversity but from having correctly identified and compositionally structured the atomic building blocks. This empirical observation, i.e., RL producing richer short-combination diversity that facilitates identification and composition, has a precise formal basis, which we now state.
> > >
> > > ---
> > >
> > > **Theory -- a formal result**
> > >
> > > > "... Theory -- a formal result showing that ... RL's estimated distribution is closer to the true one than SFT's, under what conditions, and by how much."
> > >
> > > Thank you for the constructive feedback. In light of your comment, we've included the following formal results in our theory section to formalize the intuition. Our mechanistic claim is that on the specific trace events required by our identification and composition theory, SFT can be provably non-identifiable, whereas RL with verifiable reward can provably increase their mass. We state the conditions and quantify the gap.
> > >
> > > **Identifiability-critical events.** Our conditions require observing specific trace-level configurations -- a module in a neighboring context (C.1-v), or a parent co-occurring with a child tuple (local witness). We call these *identifiability-critical events*.
> > >
> > > **Under what conditions does SFT fail?** Let $Q^\star(p,d)=\mu^\star(p)p^\star(d\mid p)$ be the true law. SFT reveals only a subset $A_S(p)$ of valid traces with visible mass $s(p)$; define $\widetilde Q(p,d)=\mu^\star(p)q_S(d\mid p)$ where $q_S(d\mid p)=p^\star(d\mid p)\mathbf{1}\lbrace d\in A_S(p)\rbrace/s(p)$. Then $Q^\star$ and $\widetilde Q$ produce identical SFT data.
> > >
> > > **By how much do they differ?** The gap is quantified by $\lVert Q^\star-\widetilde Q\rVert_1 = 2\mathbb{E}_{\mu^\star}[1-s(p)]$. Therefore, any SFT-only estimator has worst-case L1 error at least this hidden-mass term. When identifiability-critical configurations (C.1-v neighborhoods, Theorem 3.2 witnesses) lie in that hidden mass, SFT cannot recover them even with infinite data. The obstruction is non-identifiability, not optimization or finite samples.
> > >
> > > **Under what conditions does RL recover, and by how much?** For an identifiability-critical event $G$ with rollout probability $p_G(p)$ and reward gap $\Delta_G(p)=\Pr(Y=1\mid G,p)-\Pr(Y=1\mid G^c,p)$. Under verifiable reward, $\Delta_G(p)>0$ is the natural regime, since correct local configurations make reaching a correct final answer more likely. When $p_G(p)>0$ and $\Delta_G(p)>0$, Bayes' rule gives
> > > $Q_0(G\mid Y=1,p)-Q_0(G\mid p)=\frac{p_G(p)(1-p_G(p))\Delta_G(p)}{\Pr(Y=1\mid p)}>0.$
> > > RL thus creates both enrichment of identifiability-critical events among reward-positive rollouts and a contrastive signal between positive and negative traces, with policy-gradient magnitude $\mathbb E_{\mu^\star}[p_G(p)(1-p_G(p))\Delta_G(p)]$. Three factors govern RL's advantage, directly answering "by how much": (1) *censorship severity* $1-s(p)$ -- larger hidden mass makes SFT worse; (2) *reachability* $p_G(p)$, missing events must be rollout-reachable; (3) *reward informativeness* $\Delta_G(p)$, correct local configurations must correlate with final-answer reward. The distributional-support interventions in Sec 4 (removing/re-injecting skills, varying composition depth) directly test these three factors, and the results align with the predicted failure and recovery modes. We will revise to state these results explicitly and to clarify that our formal claim concerns recovery of the critical local support needed by Theorems 3.1/3.2, not an unconditional full-distribution dominance claim.
> > >
> > > We are grateful for your insights and your contribution to improving our work!

---

### Official Review · Reviewer_RjLb · 2026-03-12

**Soundness:** 3
**Presentation:** 3
**Significance:** 3
**Originality:** 3
**Overall Recommendation:** 4
**Confidence:** 4

**Summary:**

This paper explains why reinforcement learning (RL) improves LLM reasoning, especially on out-of-distribution tasks. The authors argue that RL enables compositional generalization, allowing models to recombine reusable reasoning modules (skills and routing mechanisms) in new ways.
They introduce a Hierarchical Latent Selection Model to formalize reasoning as a sequence of latent module selections. Theoretically and empirically, they show that RL’s exploration helps discover these modules and recombine them to solve new problems.
They also find that SFT learns the basic modules, while RL encourages new compositions of them, leading to stronger generalization.

**Compliance With Llm Reviewing Policy:**

Affirmed.

**Final Justification:**

I have decided to maintain my score. The authors have provided a thorough and convincing response to my initial concerns, particularly regarding the novelty of their mechanistic explanation and the practical implications of their theoretical framework.

**Key Questions For Authors:**

- Some recent work suggests that SFT expands the space of reasoning traces while RL “squeezes” or refines them [2]. How does the mechanism proposed in this paper relate to that perspective? Are these views complementary or contradictory?

- The synthetic tasks in the paper primarily follow a nested compositional structure such as $f_3(f_2(f_1(x)))$. However, many real reasoning problems may not follow such a simple nested structure—for example, problems requiring case analysis or branching. Do the authors expect the same modular-composition framework to extend to these settings?

- The work mainly provides analysis and empirical observations about RL’s role in compositional reasoning. What practical implications do the authors envision? For example: Could this framework suggest new RL objectives or curricula? Should training datasets explicitly include designed compositional structures? Are there implications for improving post-training pipelines in real LLM systems?

[2] Matsutani et al., "RL Squeezes, SFT Expands: A Comparative Study of Reasoning LLMs"

**Limitations:**

yes

**Strengths And Weaknesses:**

**Strengths**

- The paper presents an interesting perspective by viewing reasoning ability as the composition of reusable skills and routing mechanisms. This modular view of reasoning is conceptually appealing and provides a useful framework for thinking about compositional generalization in LLMs.

- The work studies the relationship between SFT and RL in post-training, which is currently a highly active research topic. The attempt to explain their behavioral differences—particularly in terms of compositional generalization—is timely and relevant.

---

**Weaknesses**

- The empirical finding that models trained with RL after SFT tend to generalize better to OOD compositions is interesting, but it appears somewhat expected. RL introduces additional training signals and computational cost, so stronger generalization may naturally follow. Moreover, prior work (e.g., “RL generalizes while SFT memorizes” [1]) has already discussed a similar phenomenon.

- Finding 2 (Composability Requires Combinational Exposure during RL) also feels somewhat intuitive. It is not surprising that models require compositional training examples to acquire compositional capabilities; without such exposure, learning how to recombine components would be difficult.

[1] Chu et al., "SFT Memorizes, RL Generalizes: A Comparative Study of Foundation Model Post-training"

---

> ### Author Rebuttal · Authors · 2026-03-30
>
> > **W1-W2:** "... RL after SFT tend to generalize better to OOD compositions is interesting, but it appears somewhat expected. ... prior work (e.g., Chu et al.) has already discussed a similar phenomenon."
> >
> > "Finding 2 ... also feels somewhat intuitive. It is not surprising that models require compositional training examples to acquire compositional capabilities ..."
>
>
> We fully agree that, at a high level, (1) RL generalizes better than SFT and (2) compositional exposure helps are consistent with intuition. Our contribution is the *mechanistic explanation* underneath.
>
>
> Prior work such as Chu et al. documents the SFT–RL gap empirically. Our paper asks *why* it arises: we formalize OOD reasoning as compositional generalization via a hierarchical latent model distinguishing skills from routing mechanisms (§2), prove conditions under which RL identifies latent modules (Theorem 3.1) and recombines them (Theorem 3.2), and validate each through controlled interventions.
>
>
> This yields new design principles hard to derive from intuition alone. Finding 4 shows *disjoint* SFT–RL composition sets maximize OOD gains – a non-obvious prescription from the theory. Theorem 3.2's witness intersection condition (Eq. 7) explains *precisely* why isolated skill re-injection fails while compositional re-injection succeeds (Finding 2, Figure 4): isolated atoms satisfy identification but lack local joint witnesses for composition.
> These conditions provide quantitative guidance for dataset construction beyond "use compositional data." We've included this discussion in our related work – thank you for the helpful comment.
>
>
> ---
>
>
> > **Q1:** "... SFT expands the space of reasoning traces while RL "squeezes" or refines them [Matsutani et al.]. ... Are these views complementary or contradictory?"
>
>
> We view the two perspectives as **complementary**. Matsutani et al. focus on the *final* distribution: SFT broadens support, RL suppresses incorrect traces. We focus on *training-time exploration*: RL's diverse rollouts enable identification and composition of reusable modules. RL explores broadly *during* learning (our focus) yet produces a concentrated distribution *after* learning (their focus). Thank you for the valuable reference, and we've included this discussion in our related work.
>
>
> ---
>
>
> > **Q2:** ... many real reasoning problems may not follow ... nested structure ... problems requiring case analysis or branching. Do the authors expect the same framework to extend ...?
>
>
> Great question – this is exactly why we introduced **routing mechanisms**, going beyond prior work (e.g., Yuan et al., 2025) which considers only nested chains. Routing mechanisms (§2) determine how intermediate information is selected, reused, and composed. In our submission, we design 10 atomic routing mechanisms covering diverse patterns. §5.4 (Finding 3) validates that RL decomposes and recomposes routing modules via the same mechanism as skills.
>
>
> ---
>
>
> > **Q3:** "... What practical implications do the authors envision? ... Are there implications for improving post-training pipelines ...?"
>
>
> Thank you for the insightful comments – our framework yields actionable prescriptions:
>
>
> 1. **SFT–RL data relationship.** Composition sets should be *disjoint* to maximize OOD generalization (Finding 4, Figure 5).
> 2. **Trace design.** SFT should cover all skills through *compositional* traces (not isolated atoms), using reusable, generic modules. RL should target novel compositions outside SFT support; the witness intersection condition (Eq. 7) specifies which interfaces need coverage.
> 3. **Re-injection.** New skills must appear within compositional contexts, not in isolation: isolated re-injection recovers an atomic skill but not its composability (Finding 2).
>
>
> We've expanded this discussion in the revision -- thank you for your helpful comments!

---

> > ### Author Rebuttal · Reviewer_RjLb · 2026-04-02
> >
> > Thank you for the detailed rebuttal and clarifications.
> >
> > While some aspects still feel somewhat intuitive at a high level, I appreciate the effort to formalize and provide a deeper explanation.
> >
> > Overall, I remain positive about this work and am happy to maintain my score.

---

> > > ### Author Response · Authors · 2026-04-02
> > >
> > > We’re very glad the rebuttal was helpful -- thank you so much for your continued support and thoughtful comments that have helped us improve our work!

---

### Official Review · Reviewer_rYNv · 2026-03-13

**Soundness:** 2
**Presentation:** 1
**Significance:** 3
**Originality:** 3
**Overall Recommendation:** 4
**Confidence:** 3

**Summary:**

This paper first formalizes reasoning as hierarchicel latent selection model, where reasoning traces could be seen as a cascade of discrete latent selection variables including atomic skills and the routing mechanisms. They provide a theory that covers identifiability of the latent structure and the requirement for witnessing local intersections for learning compositional generalization. They perform empirical investigation using a previously used synthetic task for learning compositional functions defined with atomic skills and the routing structures, and they compare model performance with the variety of setups changing SFT/RL coverage of atomic skills and compositions.

**Compliance With Llm Reviewing Policy:**

Affirmed.

**Final Justification:**

I thank the authors for their time and effort for the rebuttal. Their last update on RL vs. SFT resolved my concern and I update my score from 3 to 4.

**Key Questions For Authors:**

Most of my major questions are already mentioned in the weaknesses.
Some minor questions:
-How is “re-injection” in RL done specifically? Do you first do RL finetune and do some more RL finetuning after including the re-injecting data? What is the experimental details?
-How does your result in real models, section 4.6 (line 430-433) coherently connect to your theory and the claims in synthetic settings? What is new from already established results that SFT tends to memorize and RL more generalize in the previous literature(e.g. Chu et al. 2025)?

**Limitations:**

Yes

**Strengths And Weaknesses:**

**Strengths**

- The authors formalized the reasoning as a hierarchical latent selection model and introduced theorems about the identifiability of the latent structure and requirement for witnessing local intersections to learn ‘composition’ although, a disclaimer, that I did not go through the full proof of the theorems.

- The authors used a synthetic task to provide a controlled empirical experiments on their hypothesis.

- The topic of how models learn atomic skills and reuse and combine modules to perform unseen reasoning and its relationship to post-training recipe is an interesting and important question.

**Weaknesses**
- My main bottleneck is theory-empirics connection. Theorem 1 reads as a identifiability given the condition of the reasoning traces, and theorem 2 is requirement for witnessed intersections. They do not fully distinguish the different constraints arising from the learning algorithm (SFT / RL).  The paper explains that RL is advantageous because multiple rollouts provide more coverage of meaningful structures but it is still mostly heuristic. Currently, there is no tight parts in theory that show RL procedure increases the theory-relevant quantities, or that the observed intervention effects are mediated by them, but just relying on heuristic assumptions that multiple RL rollouts support them, which is in particularly unclear when we consider GRPO like verifiable rewards as learning signal. I think this limitation should be more clearly stated.
- The readability could be improved. While I appreciate that the authors tried to contextualizet their theories by providing some example scenarios, I found it hard to follow theoretical setup and its implication and currently a big bottleneck to see how the current theories are connected to the empirical findings that authors proposed. Furthermore, the experimental setup lacks lots of details; what architecture they use, what are detailed training setups, etc.
- A significance of some of the empirical experimental results remains unclear. All the experiments are done in a single seed. While I understand the limitation in compute budget, but some results have a small gap, and verification with more than a single seed seems to be beneficial to claim the significance of their conclusion, especially Figure 5 and Figure 7. In a similar vein, many experiments does not show the converged status, and together with the small margin at no-convergence, it is hard to commit to the current empirical results.

---

> ### Author Rebuttal · Authors · 2026-03-30
>
> **Notes**: All the figures can be found at https://anonymous.4open.science/r/res-C767/Response_to_reviewer_rYNv.pdf.
>
> ---
>
>
> > **W1:** "My main bottleneck is theory-empirics connection. ... it is still mostly heuristic. ... there is no tight parts in theory that show RL procedure increases the theory-relevant quantities ... I think this limitation should be more clearly stated."
>
>
> Thank you for the thoughtful comments. We've revised to make the theory–empirics connection clearer. SFT and RL differ in the data they can observe from the *same problem set*: RL can elicit more information. This directly determines whether our identifiability conditions can be satisfied.
>
>
> Specifically, our identifiability conditions (Condition C.1) are stated on the *true* data-generating distribution $P(\mathbf{D} \mid \mathbf{P})$. SFT and RL each yield a different *estimated* distribution. SFT observes only curated correct traces (one or few per problem). RL generates $K$ rollouts per problem, producing both successful traces and *failed rollouts*. The reward signal reveals which succeed and which fail, so RL observes a richer slice of the true distribution, and the conditions for identification become satisfiable, particularly *neighborhood coverage* (C.1-v) that requires each module to appear in enough distinct contexts.
>
>
> Let us illustrate this with an example: if `sort()` only ever appears before "take the median" in SFT data, the learner fuses the two; RL's diverse rollouts place `sort()` in other contexts, isolating it as a reusable module.
>
>
> This is consistent with established results in imitation learning (e.g., Ross et al. [R1]): positive-only imitation suffers compounding errors. SFT cannot reveal invalid module configurations, whereas RL's failed rollouts expose exactly these.
>
>
> In addition, thanks to your comment, we've included in our revision the number of compositions generated by RL vs. SFT models (please see Figure H1 in the link). As you can see, RL provides substantially more distinct compositions, confirming the theoretical prediction.
>
>
> ---
>
>
> > **W2:** "The readability could be improved. ... the experimental setup lacks lots of details ..."
>
>
> Thank you – we've made two concrete improvements:
>
>
> (1) **Theory–experiment roadmap.** We've included a new paragraph at § 4's start to clearly indicate and explain how *Theorem 3.1* is tested in §4.2 and *Theorem 3.2* is tested in §4.2–4.5.
>
>
> (2) **Experimental details.** We've included the following detail at § 5.1: SFT fine-tunes Llama-3.1-8B-Instruct for 2 epochs (lr $2 \times 10^{-5}$, batch 128). RL applies DAPO (lr $1 \times 10^{-6}$, max length 8192, temperature 1.0, batch 16, KL/entropy coefficients 0), filtering problems where all rollouts are correct or all incorrect. Other hyperparameters use veRL defaults.
>
>
> ---
>
>
> > **W3:** "All the experiments are done in a single seed. ... some results have a small gap ... many experiments does not show the converged status ..."
>
>
> Thanks to your helpful suggestion. We're running experiments in Figure 5 and Figure 7 with different seeds and longer until convergence. Current results are visualized in Figure H2-H5 in the link. As you can see, conclusions remain the same. We'll update you as soon as full results are ready – thank you for your patience!
>
>
> ---
>
>
> > **Q1:** "How is "re-injection" in RL done specifically? ... What is the experimental details?"
>
>
> All settings share the same SFT model trained on $L=3$ traces. We remove compositions involving the held-out skill $f_\star$ from the base RL corpus, then augment with re-injection traces, either isolated $f_\star$ at $L=1$ or compositions containing $f_\star$ at varying depths. RL proceeds from the SFT checkpoint on this modified corpus; no second finetuning round. We've made this clearer in the revision -- thank you for the suggestion.
>
>
> ---
>
>
> > **Q2:** "How does your result in real models ... coherently connect to your theory ...? What is new from ... Chu et al. 2025?"
>
>
> §4.6 tests the same mechanism in real models: RL-trained models recombine short, reusable skill n-grams, while SFT models rely on longer memorized sequences – exactly the signature of Theorems 3.1–3.2.
>
>
> On novelty: prior work documents the SFT–RL gap empirically; we contribute a *mechanistic explanation*. Specifically, we (i) formalize OOD reasoning as compositional generalization, (ii) distinguish skills from routing mechanisms, (iii) prove identification and composition conditions (Theorems 3.1–3.2), (iv) validate via controlled interventions, and (v) derive actionable design principles (e.g., Finding 4's SFT–RL data relations). We've revised §4.6 and related work accordingly. Thank you for your suggestion!
>
> **Reference**:
>
> [R1] Ross et al., "A Reduction of Imitation Learning and Structured Prediction to No-Regret Online Learning," AISTATS 2011.

---

> > ### Author Rebuttal · Reviewer_rYNv · 2026-04-03
> >
> > I appreciate the authors for their time and effort on the rebuttal.
> > I thank the authors for extending their experiments and improving the manuscript.
> >
> > Nevertheless, my major concern still remains about the loose end of theory - experiment connection. The authors answered,
> > "SFT and RL differ in the data they can observe from the same problem set: RL can elicit more information. This directly determines whether our identifiability conditions can be satisfied.... RL observes a richer slice of the true distribution", and I do not think the current theory includes formal justification of the first critical part, "RL can elicit more information". The authors provide examples on sort(), but again, this is still a heuristic example rather than a formal verification.

---

> > > ### Author Response · Authors · 2026-04-04
> > >
> > > (**Updated on April 7**) Again, thank you for this insightful feedback and your patience! In light of your feedback, we've included the following results in our theory section to pin down exactly what "RL can elicit more information" means. We show that under the same prompt distribution, SFT and RL expose different *within-prompt trace supports*, and this difference determines whether our identifiability conditions can be met.
> > >
> > > **Identifiability-critical events.** Our conditions require the learner to observe specific trace-level configurations -- a module in a particular neighboring context (Condition C.1-v), or a parent value co-occurring with a specific child tuple (local witness in Theorem 3.2). We call these *identifiability-critical events*; the key question is whether a given training regime exposes enough of them.
> > >
> > > **SFT impossibility.** Let $Q^\star(p,d)=\mu^\star(p)p^\star(d\mid p)$ be the true prompt-trace law. SFT reveals only a canonical subset $A_S(p)$ of valid traces, with visible conditional $q_S(d\mid p)=p^\star(d\mid p)\mathbf{1}\{d\in A_S(p)\}/s(p)$, where $s(p)$ is the visible mass. Define $\widetilde Q(p,d)=\mu^\star(p)q_S(d\mid p)$. Then $Q^\star$ and $\widetilde Q$ generate exactly the same SFT data, yet $\lVert Q^\star-\widetilde Q\rVert_1=2\,\mathbb{E}_{\mu^\star}[1-s(p)]$. Hence any configuration in the hidden mass, e.g., the alternative neighborhoods required by Condition C.1-v or the local witnesses in Theorem 3.2, is formally unidentifiable from SFT alone, even with infinite data. The obstruction is non-identifiability, not optimization or finite samples.
> > >
> > > *Proof sketch.* Under $Q^\star$, SFT reveals traces as $q_S$; under $\widetilde Q$, the full law *is* $q_S$, so SFT data is identical in both worlds and identifiability conditions can fail under $\widetilde Q$ while remaining unfalsifiable.
> > >
> > > **RL recovery.** RL samples rollouts rather than replaying only $A_S(p)$. For any identifiability-critical event $G$, let $p_G(p)$ be its rollout probability -- the chance a random rollout under prompt $p$ contains $G$ -- and $\Delta_G(p)=\Pr(Y=1\mid G,p)-\Pr(Y=1\mid G^c,p)$ the within-prompt reward gap. Under verifiable reward, $\Delta_G(p)>0$ is the natural regime: identifiability-critical events are correct local configurations (right module in right context), and traces using them are more likely to reach a correct final answer. When $p_G(p)>0$ and $\Delta_G(p)>0$, Bayes' rule gives
> > > $Q_0(G\mid Y=1,p)-Q_0(G\mid p)=\frac{p_G(p)(1-p_G(p))\Delta_G(p)}{\Pr(Y=1\mid p)}>0,$
> > > so RL creates three distinct effects: enrichment of $G$ among reward-positive rollouts, a contrastive signal between positive and negative traces, and a local ascent direction for expected reward, with policy-gradient magnitude $\mathbb E_{\mu^\star}[p_G(p)(1-p_G(p))\Delta_G(p)]$. On the same prompt set, RL surfaces and amplifies trace configurations that SFT never reveals. That is the formal content behind "RL can elicit more information": richer observation of the trace distribution, not a change in prompts.
> > >
> > > *Proof sketch.* Enrichment and contrastive signal follow from Bayes' rule within each prompt; the ascent direction follows from the policy-gradient covariance identity.
> > >
> > > **Theory-experiment connection.** These results predict specific experimental signatures: (i) Sec. 4.2 -- RL improves $L=1$ atomic recovery from $L=3$-only training, because RL surfaces hidden within-prompt configurations; (ii) Secs. 4.3-4.4 -- removing a skill or router from RL hurts exactly those compositions requiring it, because removing reachable events $G$ breaks recovery; (iii) re-injecting composed traces containing a held-out skill restores performance but isolated atom re-injection does not, because the witness condition (Theorem 3.2) requires joint parent-child events.
> > >
> > > In summary, three factors govern when RL surpasses SFT: (1) *censorship severity* -- larger hidden SFT mass $1-s(p)$ widens non-identifiability; (2) *reachability* -- missing events must have $p_G(p)>0$; if unreachable, neither random exploration nor RL helps; (3) *reward informativeness* -- $\Delta_G(p)>0$ is needed for RL amplification; when $\Delta_G(p)=0$, RL reduces to random exploration for that event. When all three align, RL has a provable advantage over SFT.
> > >
> > > Thank you for your valuable suggestion to have strengthened our work!

---

### Decision · Program_Chairs · 2026-04-30

**Decision:**

Accept (regular)

**Comment:**

This paper formalizes why RL post-training improves LLM reasoning on out-of-distribution tasks, arguing that RL enables compositional generalization by identifying and recombining reusable atomic modules (skills and routing mechanisms). The authors introduce a Hierarchical Latent Selection Model, prove identifiability and composition conditions, and validate their theory through controlled synthetic experiments and real-model analysis. All four reviewers gave scores of 4 (Weak Accept), recognizing the timeliness and promise of the research direction. The paper addresses a significant question with a novel theoretical lens, the controlled experiments are well-designed. I recommend weak accept, and strongly encourage the authors to integrate the rebuttal-stage formal results and additional experiments into the final version with care.